# A parsimonious transport model of emerging contaminants at the river network scale

Elena Diamantini[1], Stefano Mallucci[2], and Alberto Bellin[1]

[1]Department of Civil, Environmental and Mechanical Engineering, University of Trento, via Mesiano 77, 38123 Trento (Italy)
[2]C3A - Center Agriculture Food Environment, University of Trento / Fondazione Edmund Mach, via Edmund Mach 1, 38010 San Michele all'Adige (Italy)

**Correspondence:** Elena Diamantini (elena.diamantini@gmail.com)

**Abstract.** Waters released from wastewater treatment plants (WWTPs) represent a relevant source of pharmaceuticals and personal care products to the aquatic environment, since many of them are not effectively removed by the treatment systems. The consumption of these products increased in the last decades and concerns have consequently risen about their possible adverse effects on the freshwater ecosystem. In this study, we present a simple, yet effective, analytical model of transport of contaminants released in surface waters by WWTPs. Transport of dissolved species is modeled by solving the Advection-Dispersion-Reaction Equation (ADRE) along the river network by using a Lagrangian approach. We applied this model to concentration data of five pharmaceuticals: diclofenac, ketoprofen, clarithromycin, sulfamethoxazole and irbesartan collected during two field campaigns, conducted in February and July 2015 in the Adige river, North-East of Italy. The model showed a good agreement with measurements and the successive application at the monthly time scale highlighted significant variations of the load due to the interplay between streamflow seasonality and variation of the anthropogenic pressure, chiefly due to the variability of touristic fluxes. Since the data required by the model are widely available, our model is suitable to large-scale applications.

## 1 Introduction

The presence of pharmaceuticals and personal care products (PPCPs) in the environment raises growing concerns because of their potential harmful effects on humans and freshwater ecosystems (Ebele et al., 2017). Despite these substances are ubiquitous in populated areas and detected in fresh waters with concentrations ranging from nanograms to micrograms per litre (see e.g., Table 1), a regular monitoring activity by Environmental Agencies is not yet enforced by regulations at the European level (Heberer, 2002; Ellis, 2006; Kuster et al., 2008; Acuña et al., 2015; Rice and Westerhoff, 2017). The main entry route of PPCPs into the aquatic environment is through the water discharged by Waste Water Treatment Plants (WWTPs), whose removal efficiency varies in dependence of the type of contaminant and the treatment technology (Halling-Sørensen et al., 1998; Rivera-Utrilla et al., 2013; Petrovic et al., 2016). The ubiquitous presence of PPCPs in freshwaters is due to the rise of urban population and the introduction of new products in the market, given the escalating request by human population and for livestock breeding. Persistence of PPCPs in freshwater varies from a few days to years, depending on both environmental conditions and characteristics of the compound. In addition, their concentration downstream of the WWTPs may change

significantly as an effect of dilution and environmental conditions, chiefly solar irradiation and water temperature. Situations of pseudo-persistence of supposedly rapidly degrading PPCPs due to multiple release have also been observed (Ebele et al., 2017). Since PPCPs are designed to exert physiological effects at low dosage, possible adverse consequences on humans and biota have become an issue of increasing concern (Heron and Pickering, 2003; Schwab et al., 2005; Boxall et al., 2012): disruption of human endocrine functions, developmental defects in fish and other organisms, alterations in the survival, growth, and reproduction of several species, and the promotion of antibiotic resistance are just a few examples of adverse effects requiring investigation (see e.g., Brooks et al., 2009; Corcoran et al., 2010; Hemond and Fechner, 2014; Ebele et al., 2017). As a first response to these concerns, the European Union emanated the Directive 2013/139/EU (Council of European Union, 2013) which identifies priority substances that might represent a potential risk and defines environmental quality standards.

Specific modeling approaches have been proposed with the objective to evaluate the propagation of PPCPs in rivers (see e.g., Scheytt et al., 2006; Osorio et al., 2012; Vione et al., 2018) all sharing the conceptual framework of GREAT-ER (Geography-Referenced Regional Exposure Assessment Tool for European Rivers) (Feijtel et al., 1997) and PhATE (Pharmaceutical Assessment and Transport Evaluation) (Anderson et al., 2004). Both are GIS-based models and take into account the decay of the species along the river in a simplified manner by considering a representative water discharge and therefore neglecting changes of dilution due to its variability in time. PhATE computes the total load upstream the point of interest and then estimates the concentration of a target compound by dividing it by a representative water discharge (Anderson et al., 2004). The model is composed of two modules: the exposure module, which estimates environmental concentrations, and the human health effect module, which is intended for risk assessment (Aldekoa et al., 2015). The effect of transport along the river network is therefore neglected and the water discharge is typically selected as representative of low flow conditions such that seasonal variations cannot be assessed. The WWTP loads are estimated by multiplying the total compound consumption, given as the product of the per capita consumption and the served population, by two reducing factors taking into account the fraction of the compound metabolized by the human organism and that removed by the WWTP. GREAT-ER was developed for applications in large river basins at the pan-European scale (Boeije et al., 1997; Koormann et al., 2006), but it has been applied also for environmental risk assessment (Kehrein et al., 2015). The river network is divided into connected segments, each one receiving the load from upstream and from both the WWTPs and the industrial sewage systems directly connected to it. The last version of the software includes a uniformly distributed injection along the segments (Kehrein et al., 2015). The model assumes stationary (constant in time) emissions such that residence time is relevant only if decay is considered. Several types of decay are included, all lumped in a first order kinetic with the residence time estimated as the ratio between the length of the segment and a reference stationary flow velocity. Similarly to PhATE, and consistently with the hypothesis of stationary release, a single water discharge representative of low flow conditions is considered to obtain concentrations from the estimated mass flux. Stationarity of emissions and the assumption of a constant and deterministic residence time does not allow to estimate the seasonal variability of PPCPs concentrations at the selected locations. However, recognizing that uncertainty plagues parameters selection, and in an attempt to evaluate its propagation to the concentration estimates, the developers of GREAT-ER included a Monte Carlo procedure to evaluate parametric uncertainty under the assumption that the parameters are normally distributed independent random variables with given means and variances.

In the present work we propose a new modeling framework which includes hydrodynamic processes and dilution occurring along the river network in a simplified, yet rigorous, manner while keeping the model parsimonious in term of number of parameters. Differently from existing models, our approach takes into account flow variations along the path from the source to the point of interest, by assuming that water discharge changes at the nodes of the network while remaining constant along the edges (streams) connecting the nodes. In doing that water discharge, and therefore stream velocity, varies stepwise along the path from the source to the point of interest and dilution is included by performing mass balance at the nodes of the network. Moreover, multiple sources are addressed in a rigorous manner by taking advantage of the linearity of the transport equation simply by adding all the contributions after their transfer to the control section (see Sect. 2). Our model removes the assumption of stationarity in both emissions and streamflow. In the present work transient flow conditions are modeled as a succession of stationary flows representative of seasonal variability, under the hypothesis that the residence time along the edge is smaller than the characteristic time of flow variations. This hypothesis can be removed at the price of a higher model complexity, which is not always justified, particularly when the objective of the analysis is the estimation of the seasonal loads, as in the present work and in most applications alike. Including variability of flow and contaminant loads, though in a simplified manner, is crucial when both water discharge and populations vary in time, the latter due to touristic fluxes, for example. Indeed, the importance of seasonality in PPCPs consumption and streamflow may be limited in large pan-European catchments (in particular the former) but becomes more influent as the size of the catchment reduces, especially in the Alpine region where touristic fluxes cause relevant seasonal variations of the population. The effects of the above variabilities have been scarcely investigated (see e.g., Alder et al., 2010), since very few studies have analyzed temporal and seasonal variations in both concentrations and overall attenuation of PPCPs loads (Loraine and Pettigrove, 2006; Robinson et al., 2007; Daneshvar et al., 2010; Aldekoa et al., 2015). Considering these variabilities requires data on streamflow and PPCPs emissions at the selected time scale, typically the daily or the monthly scale. Streamflow may be obtained from recorded data or from hydrological modeling. Similarly to GREAT-ER, all decay processes are lumped in a single first-order decay rate, and sorption by sediments is included through a linear equilibrium isotherm. Both the decay rate and the partition coefficient are temperature dependent through the Arrhenius law.

To summarize, we propose a new parsimonious, in terms of parameters, in-stream transport model which includes the concurrent effects of dilution, dispersion and decay of PPCPs in surface waters. Its strength is in the parametrization of the releases as a function of human resident population and touristic fluxes, the latter varying seasonally, both considered as a proxy of the sewage effluents. Human population and touristic fluxes data are widely available and this makes the model applicable in a variety of situations, despite the lack of systematic data on the contamination by PPCPs. Moreover, our model can be easily coupled to existing hydro-climatological models providing streamflow and water temperatures in the catchment of interest and is consistent with the general framework developed by Botter et al. (2011) under the hypothesis that transient flow can be represented as a superimposition of stationary flow fields.

The model is presented in Sect. 2, whereas Sect. 3 describes the Adige river basin and the data used for illustrating model's application. Section 4 presents the inference of model parameters and Sect. 5 discusses the simulations performed to estimate the seasonal loads in the Adige catchment. Finally, a discussion of the main findings and the conclusions are presented in Sect. 6.

**Table 1.** Concentrations of the 5 selected pharmaceuticals recorded in surficial waters worldwide.

| Diclofenac | | |
|---|---|---|
| *Country* | *Concentration [ng/l]* | *Reference* |
| U.S. | 21 - 34 | Mohapatra et al. (2016) |
| Switzerland | 99 | Tixier et al. (2003) |
| Canada | 26 - 194 | Metcalfe et al. (2003) |
| France | 290 - 410 | Ferrari et al. (2004) |
| Germany | 420 - 2100 | Ferrari et al. (2004) |
| Sweden | 10 - 120 | Bendz et al. (2005) |
| Spain | 0.25-280 | Aldekoa et al. (2013) |
| India | 1.41-41.3 | Sharma et al. (2019) |
| Malaysia | 1.11-4.92 | Praveena et al. (2018) |
| Northern Antarctic Peninsula | 7761 | González-Alonso et al. (2017) |
| Spain | 89.53–176.78 | López-Serna et al. (2012) |
| Mexico | 258–1398 | Rivera-Jaimes et al. (2018) |

| Ketoprofen | | |
|---|---|---|
| *Country* | *Concentration [ng/l]* | *Reference* |
| Switzerland | 180 | Tixier et al. (2003) |
| Australia | <10 | Scott et al. (2014) |
| Canada | 12 - 50 | Metcalfe et al. (2003) |
| Sweden | 10 - 70 | Bendz et al. (2005) |
| India | 2.71-107 | Sharma et al. (2019) |

| Clarithromycin | | |
|---|---|---|
| *Country* | *Concentration [ng/l]* | *Reference* |
| U.S. | 48 - 66 | Mohapatra et al. (2016) |
| South Korea | 49 - 443 | Kim et al. (2009) |
| Italy | 8.3 - 20.3 | Zuccato et al. (2005) |
| Japan | 232 | Arizono (2006) |
| Germany | 210 | Ternes et al. (2007) |
| China | 9.9 | Asghar et al. (2018) |

| Irbesartan | | |
|---|---|---|
| *Country* | *Concentration [ng/l]* | *Reference* |
| Slovene | 0.2 – 9.3 | Klančar et al. (2018) |
| China | 18 | Asghar et al. (2018) |

| Sulfamethoxazole | | |
|---|---|---|
| *Country* | *Concentration [ng/l]* | *Reference* |
| U.S. | 313 - 342 | Mohapatra et al. (2016) |
| India | 357 | Mohapatra et al. (2016) |
| Australia | <5 | Scott et al. (2014) |
| France | 70 - 90 | Ferrari et al. (2004) |
| Germany | 480 - 2000 | Ferrari et al. (2004) |
| Sweden | 0 - 10 | Bendz et al. (2005) |
| Malaysia | 19.3 - 75.5 | Praveena et al. (2018) |
| China | 2.5 | Asghar et al. (2018) |

## 2  The Model

The building block of our modeling approach is the solution of the one-dimensional Advection Dispersion Reaction Equation (ADRE) within a channel (stream) connecting two nodes of the river network (Bachmat and Bear, 1964):

$$(1+K_d)\frac{\partial C}{\partial t} + v\,\frac{\partial C}{\partial x} = \alpha_L\,v\,\frac{\partial^2 C}{\partial x^2} + r \tag{1}$$

where $C$ $[\mathrm{M\,L^{-3}}]$ is the solute concentration, $x$ $[\mathrm{L}]$ is the Lagrangian coordinate measured along the channel, $t$ $[\mathrm{T}]$ is time, $v$ $[\mathrm{L\,T^{-1}}]$ is the mean velocity, $\alpha_L$ $[\mathrm{L}]$ is the local dispersivity, $K_d$ $[-]$ is the partition coefficient of the linear equilibrium isotherm representing sorption to the sediments, and $r$ $[\mathrm{M\,L^{-3}\,T^{-1}}]$ is the sink/source term representing the decay due to bio-geochemical reactions occurring in the liquid phase. The solute decays according to a first-order irreversible reaction $r = -k\,C$, where $k$ $[\mathrm{T^{-1}}]$ is the reaction rate lumping all the decay mechanisms occurring in the liquid phase. By introducing the following transformation: $C(x,t) = \tilde{C}(x,t)\,e^{-k\,t}$, Eq. (1) reduces to the classical Advection Dispersion Equation (ADE) in the transformed concentration $\tilde{C}$:

$$(1+K_d)\frac{\partial \tilde{C}}{\partial t} + v\,\frac{\partial \tilde{C}}{\partial x} = \alpha_L\,v\,\frac{\partial^2 \tilde{C}}{\partial x^2} \tag{2}$$

The model (Eq. 1) assumes that the velocity is steady state, but it can be used to simulate representative states of a slowly varying flow approximated as the superimposition of a sequence of steady state velocity fields. This is acceptable if the characteristic time of water discharge variations is larger than the residence time within the channel. Considering the typical lengths of the channels composing a river network and the time scales at which the PPCPs loads are available, time variability can be captured at daily or larger time scales. However, also in steady state conditions water discharge, and therefore flow velocity, changes along the river network. The corresponding spatial variability can be captured by means of the following power law expression:

$$v = \Phi(A)\,Q^{\Psi(A)} \tag{3}$$

where $Q$ is the water discharge. Equation (3) was proposed by Dodov and Foufoula-Georgiou (2004) for the velocity $v_p$ corresponding to the p-quantile, $Q_p$, of the water discharge as a generalization of the following power law expression: $v_p \propto Q_p^m$, introduced in the pioneering work of Leopold and Maddock (1953), who noticed that the exponent $m$ varies in dependence of the contributing area $A$ $[\mathrm{km^2}]$. In addition, $\Phi$ and $\Psi$ are time-invariant scaling coefficients in agreement with the analysis of Dodov and Foufoula-Georgiou (2004) who showed that they are independent of the chosen quantile. The expressions of $\Phi$ and $\Psi$, provided by Dodov and Foufoula-Georgiou (2004) for a representative dataset of 85 gauging stations in Kansas and Oklahoma, are reproduced in the Appendix A.

Equation (2) should be complemented with suitable initial and boundary conditions. The initial conditions are of zero concentration $C(x,0) = 0$ and zero absorbed concentration $C^*(x,0) = 0$ along the channel. A suitable upstream boundary

condition, mimicking the typical release condition at the WWTPs and industrial sewage systems is of continuous mass flux injection $\dot{M}(t)$ [M T$^{-1}$] at the position $x_0$ along the channel. In addition, we assume that the channel is semi-indefinite with the boundary condition of zero flux concentration, $C_F(x,t) = \tilde{C}(x,t) - \alpha_L \partial \tilde{C}(x,t)/\partial x = 0$ at $x \to \infty$. This condition is equivalent to assuming that the downstream boundary condition at $x = L$ does not affect mass flux.

Owing to the linearity of Eq. (2), the flux concentration $C_F$ of the solute at a given position $x$ along the channel assumes the following expression:

$$C_F(x,t) = \int_0^t C_{F,in}(t_0)\, g_M(x,t-t_0)\, dt_0 \tag{4}$$

where $C_{F,in} = \dot{M}(t)/Q(t)$ is the flux concentration at the injection point $x = x_0$, under the assumption that the release mass rate is $\dot{M}$ and that mixing with stream water occurs instantaneously. In Eq. (4) the transfer function assumes the following

form:

$$g_M(x,t) = g(x,t)\, e^{-kt} \tag{5}$$

with

$$g(x,t) = \frac{x - x_0}{\sqrt{4\pi\, \alpha_L\, v_R\, t^3}} \exp\left\{ -\frac{[x - x_0 - v_R t]^2}{4\,\alpha_L\, v_R\, t} \right\} \tag{6}$$

being the solution of the Eq. (2) for an instantaneous mass injection such that:

$c_F(x_0,t) = \delta(t) \tag{7}$

where $c_F = \dot{M}/M = C_{F,in}/(M/Q)$ [T$^{-1}$] is the flux concentration for a unit ratio between the total injected mass and water discharge and $\delta(\cdot)$ [T$^{-1}$] is the Dirac delta function. Equation (6) is the classical solution discussed in Kreft and Zuber (1978, Eq. 11) for both injection and detection in flux and unitary ratio $M/Q$.

## 2.1   Extension to the river network

Let us consider the generic network represented in Fig. (1). For an injection occurring at the position $x_{0,i}$ along the channel number $i$ the solute follows this path:

$$(i,1) \to (i,2) \to (i,3) \to \ldots\ldots(i,j) \to \ldots..(i,n_i) \tag{8}$$

where $(i,j)$ indicates the $j-$th channel in the ordered sequence of channels connecting the injection point to the control section $CS$, and $n_i$ is the total number of elements in the sequence. In the presence of lakes or reservoirs, corresponding elements

should be added to the ordered sequence. The input signal at $x = x_{0,i}$ is transferred to the end of the channel $i \equiv (i,1)$ by means of the expression (4), which can be rewritten as follows:

$$C_F(L_{(i,1)}, t) = C^{(i)}_{F,in}(t) * g_M(L_{(i,1)} - x_{0,i}, t) \tag{9}$$

where $C^{(i)}_{F,in}$ is the concentration flux of the solute released at the coordinate $x_{0,i}$ along the channel $i$ with length $L_{(i,1)}$ and the symbol $*$ indicates the convolution integral of Eq. (4).

The signal arriving from the channel $(i,1)$ is first modified to take into account the effect of dilution of merging channel(s) (i.e. the channel $l$ in the Fig. 1) and then it is propagated to the end of the following channel (i.e. the channel $(i,2)$). The same procedure is repeated for all the channels composing the path from the source to the control section. At the $j$−th channel in the sequence the convolution assumes the following form:

$$C_F(L_{(i,j)}, t) = \left\{ \frac{Q_{(i,j-1)}(t)}{Q_{(i,j)}(t)} C_F(L_{(i,j-1)}, t) \right\} * g_M(L_{(i,j)}, t); \quad j = 2, ....., n_i \tag{10}$$

For the element lake or reservoir, a similar expression can be written with $g_M(L_{(i,j)}, t)$ replaced by the function $g_M(s_k, t)$, representing the residence time distribution within this element (here $s_k$ identifies the storage element encountered along the path). For simplicity, in the following with the term lake we will indicate both natural lakes and reservoirs. Finally, owing to linearity of the transport processes, the flux concentrations $C_F(L_{(i,n_i)})$, $i = 1, ...N$, propagated from the $N$ sources within the catchment are summed up to obtain the total concentration flux at the control section $CS$:

$$C_{F,S}(t) = \sum_{i=1}^{N} C_F(L_{(i,n_i)}, t) \tag{11}$$

Under the additional assumption that $\frac{Q_{(i,j-1)}(t)}{Q_{(i,j)}(t)} \simeq \frac{A_{(i,j-1)}}{A_{(i,j)}}$, where $A_{(i,j)}$ is the contributing area to the $j$−th channel, the sequence of convolutions assumes the following form:

$$C_F(L_{(i,n_i)}, t) = \frac{A_{(i,1)}}{A_{(i,n_i)}} g_M(L_{(i,1)} - x_{0,i}, t) * g_M(L_{(i,2)}, t) * ..... * g_M(L_{(i,n_i)}, t) * g_M(s_1, t) * ..... * g_M(s_{n_s}, t) * C^{(i)}_{F,in}(t) \tag{12}$$

with $A_{(i,n_i)} = A_S$, the contributing area at the control section, and $s_k$, $k = 1, ...n_s$, that identifies the $k$-th of $n_s$ lake elements belonging to the path. The probability density function (pdf) of the lake element depends on the nature of mixing occurring in the lake and varies between an exponential pdf, in case of full mixing, and a Dirac delta distribution, in case of plug flow (Botter et al., 2005, 2011).

At the release point $x_{0,i}$ the flux concentration can be expressed as follows: $C^{(i)}_{F,in}(t) = \frac{\dot{M}_i(t)}{Q_{(i,1)}(t)}$, where $\dot{M}$ indicates the released mass flux from the WWTP and $Q_{(i,1)} = Q_i$ is the stream water discharge at the release point. The released mass flux

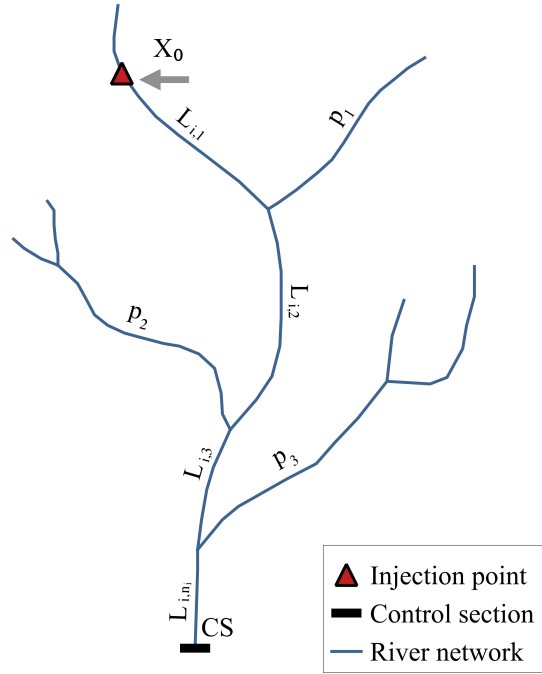

**Figure 1.** Sketch of the network with indicated the streams labels, a release point at the Lagrangian coordinate $x_{0,i}$ along the $i-$th stream, and the control section (CS). The distance between the i-th release and the Control Section is defined as $\sum_{j=1}^{n_i} L_{i,j}$. Along this path, tributaries ($p_1$, $p_2$ and $p_3$ in figure) contribute, thereby causing dilution.

from the $i-$th WWTP is given by the product of the unitary released mass flux (i.e. the mass flux released per person) $\gamma_i$ $[\mathrm{M\,T^{-1}}]$ and the population $P_i(t)$ served by the WWTP: $\dot{M}_i(t) = \gamma_i P_i(t)$. The unitary mass flux is given by:

$$\gamma_i = \frac{\alpha_i\, D_i\, \beta_i\, (1 - f_i)}{\Delta T} \tag{13}$$

where $\alpha_i$ $[-]$ is the assimilation factor, corresponding to the fraction of daily dose $D_i$ $[\mathrm{M\,T^{-1}}]$ per person that is released by
5   the human body, $\beta_i$ $[-]$ is the percentage of usage of the targeted active principle, $f_i < 1$ is the decay factor of the WWTP and $\Delta T$ is the transformation factor of time to make the units congruent with the time step used in the model.

Notice that the model is based on a segmentation of the path from the source to the control section. Therefore, diffused contributions can be evaluated at the level of the sub-catchment and treated as a point source located at the middle of the channel draining the sub-catchment. The length of the channels composing the river network, and therefore the size of the
10   sub-catchments can be varied, according to the desired level of detail (see e.g. Rodriguez-Iturbe and Rinaldo, 1997). Hence,

the maximum detail with which the spatial variability of the diffused contribution is reproduced can be controlled by the modeler simply by changing the density of the network. Notice that this has also an effect on the minimum time scale at which variability of the flow field can be captured, since the channel length influences the residence time of the stream unit. In this study, sources of diffused origin are not relevant since the region uses separate sewer systems, which eliminate sewer's overflow and the possible input from manure cannot be evaluated with the available information.

The classical study by Rinaldo et al. (1991) showed that geomorphological dispersion acting at the network scale overwhelms local dispersion in shaping the hydrological response of a catchment. The same assumption can be introduced in our model, after numerical verification, in which dilution due to the progressive increase of water discharge as the solute moves downstream rapidly overwhelms dilution due to local dispersion acting at the level of the channel, which can be neglected by assuming $\alpha_L \to 0$. Under this condition, the transfer function of the channel (i.e. Eq. 5), with $g$ provided by Eq. (6), reduces to:

$$g_M(x, t - t_0) = \delta\left[x - x_0 - v_R\left(t - t_0\right)\right]\exp\left[-k\left(t - t_0\right)\right] \tag{14}$$

where $\delta\left[T^{-1}\right]$ is the Dirac Delta distribution. Neglecting $\alpha_L$ has the advantage of reducing by one the number of parameters that should be inferred from the data, thereby diminishing the risk of over-parameterization, when, as often occurs, concentration data are scarce. In the absence of lakes the substitution of Eq. (14) into Eq. (12) leads, to the following expression of the flux concentration:

$$C_F(L_{(i,n_i)}, t) = \frac{A_{(i,1)}}{A_{(i,n_i)}} C_{F,in}^{(i)}\left(t - \sum_{j=1}^{n_i}\tau_{i,j}\right)\exp\left[-k\sum_{j=1}^{n_i}\tau_{i,j}\right] \tag{15}$$

If a lake is encountered along the path and its functioning can be represented as a plug-flow such that $g_M(s_k, t) = \delta(t - \tau_{s_k})$, Eq. (15) should be generalized by adding the residence time $\tau_{s_k}$ (and that of the other lakes encountered along the path) to the channels residence times $\tau_{i,j}$. If the hypothesis of plug-flow does not hold, the pdfs of all the lakes encountered along the path should be convoluted to the Eq. (15).

The travel time $\tau_{i,j}$ of the $i-$th channel along the path (i.e. Eq. 8) assumes the following expression:

$$\tau_{i,j} = \frac{L_{(i,j)} - \delta_{1j}x_{0,i}}{v_{0,R}(A_{(i,j)})} \tag{16}$$

where the retarded velocity is given by

$$v_{0,R}(A_{(i,j)}) = \frac{1}{1 + K_d}\,\Phi(A_{(i,j)})\left(q\,A_{(i,j)}\right)^{\Psi(A_{(i,j)})} \tag{17}$$

In Eq. (17), $q\,[\mathrm{L\,T^{-1}}]$ is the specific water discharge (i.e. the water discharge per unit contributing area is considered constant through the catchment). In Eq. (16) $\delta_{1j}$ is the Kronecker delta, which is equal to $1$ when $j = 1$ and zero otherwise.

Finally, according to the Arrhenius law, the coefficient of decay $k$ assumes the following expression (Arrhenius, 1889):

$$k = A \exp \left[ -\frac{E_A}{\mathcal{R}\theta} \right] \tag{18}$$

where $A$ [s$^{-1}$] is the frequency factor, $E_A$ [kJ mol$^{-1}$] is the activation energy, $\mathcal{R} = 8.314 * 10^{-3}$ kJ K$^{-1}$mol$^{-1}$ is the gas constant and $\theta$ [K] is the water temperature. Notice that $v_{0,R}$ changes along the river network according to the contributing area.

## 3 Materials and methods

### 3.1 The Adige river basin

We applied our model to the Adige, a large Alpine river in the North-Eastern Italy, with the parameters inferred by means of inverse modeling applied to one of its main tributaries: the Noce river. The Adige catchment area is of 12,100 km$^2$ (Fig. 2) with the large majority of the basin (91%) belonging to the Trentino-Alto Adige region. The main stem has a length of 409 km from the spring to the estuary in the Adriatic sea. Along its course the river receives the contributions of Passirio, Isarco, Rienza, Noce, Avisio, Fersina and Leno. Streamflow is characterized by a first maximum in spring, due to snowmelting, and a second one in autumn caused by cyclonic storms. Climate is typically Alpine and characterized by dry winters, snow and glacier-melt in spring, and humid summers and autumns (Lutz et al., 2016). The Noce rises from the reliefs of the Ortles-Cevedale and Adamello-Presanella groups and flows first to East, then around its middle course it turns to South-East and enters the Adige river close to the town of Mezzolombardo, North of the city of Trento (Fig. 2). Its total contributing area is of $1,367$ km$^2$ and the main stem is $82$ km long (Majone et al., 2016).

The Noce river is exploited for hydropower production with 4 reservoirs, two in the upper course (Careser and Pian Palù) and two in the middle course (S. Giustina and Mollaro). Careser and Pian Palù are in headwaters with no WWTPs upstream and therefore they enter in the model only with their effect on the water discharge. The other two are downstream of a few WWTPs (see Fig. 2 and Table B1 in the Appendix B) and therefore their effect on the residence time should be included. The Mollaro reservoir is just downstream of the S. Giustina reservoir and since no release points are in between, we merged them in a single equivalent reservoir. A recent publication of the Hydrological Observatory of the Province of Trento (2007) shows that in the period 2001-2005 the average operational volume stored in the S. Giustina reservoir was of $120.89 \times 10^6$ m$^3$. In the same period the mean water discharge was $25.8$ m$^3$ s$^{-1}$, thereby leading to a mean residence time of $\tau_{s_1} = \overline{V}/\overline{Q} = 53.7$ days. Mollaro has a little storage volume compared to that of S. Giustina. At the maximum storage (i.e. $0.860 \times 10^6$ m$^3$) the mean residence time is of $\tau_{s_2} = 0.38$ days, which summed to the mean residence time of S. Giustina leads to a total residence time of the two reservoirs of $\tau_s = \tau_{s_1} + \tau_{s_2} = 54$ days. Notice that the storage of Mollaro has been considered constant because of its small volume which allows very little flexibility for storing the water released from the S. Giustina reservoir (the S. Giustina reservoir feeds the Taio power plant which release point is just upstream of the Mollaro reservoir) and counts only for a small fraction of the total residence time of the two reservoirs. In this situation the water coming from S. Giustina and the small

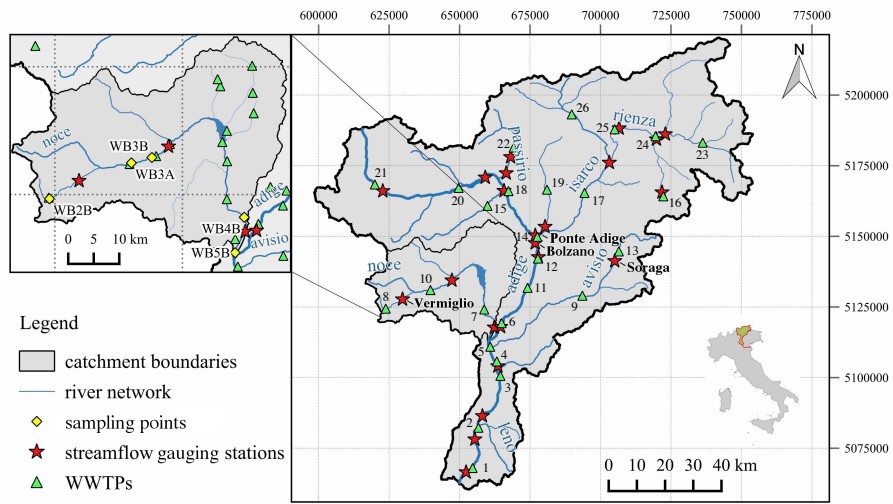

**Figure 2.** Map of the Adige river basin and its main tributaries with a zoom on the Noce river basin (upper left panel). Green triangles represent the main WWTPs whereas red stars indicate the gauging stations. The gauging stations used as control sections are identified with their names (i.e. Soraga, Vermiglio, Ponte Adige and Bronzolo). Only for the Noce basin, yellow diamonds shows the sampled locations during the two sampling campaigns of February and July, 2015.

catchment between the two reservoirs is stored for a very short time in the Mollaro reservoir with respect to the residence time of S. Giustina, such that fluctuations of its storage volume are not influencing significantly $\tau_s$.

### 3.2 Meteorological, hydrological and chemical data

Stream water of the Noce was sampled in two sampling campaigns performed respectively on February 15-17, 2015 and July
3-5, 2015 at the following sites (see Fig. 2): Tonale pass, immediately downstream of the WWTP serving a large ski area (WB2B), two sites in Mezzana, immediately downstream of the WWTP serving the middle Sole valley (WB3A and WB3B) and in two sites in the town of Mezzocorona, lower Non valley, immediately upstream and downstream of the restitution of the Mezzocorona power plant (WB4B and WB5B), respectively. The Mezzocorona power plant is fed by the water of the Mollaro reservoir (see Sect. 3). Details on sampling procedures, sampling locations, and analyses performed are provided in the work
by Mandaric et al. (2017). Table 2 reports streamflow and water temperature data measured during the two campaigns at the five sampling sites. Notice that both campaigns were performed in dry conditions (i.e. in the absence of precipitation during the samplings). These two periods were selected such as to capture extreme conditions in the catchment. Winter is the main touristic season with a large number of tourists hosted in hotels and houses in the ski area and along the Sole valley, while streamflow is at the annual minimum. On the other hand, summer is characterized by lower, yet significant, touristic presences
and high streamflow due to snow melting. In the winter campaign, 36 out of the 80 investigated pharmaceuticals were detected in water samples with concentrations above their respective Limit Of Quantification (LOQ) whereas in the summer campaign, this number reduced to 15, and with concentrations mostly lower than in winter (Mandaric et al., 2017). The quality of the

**Table 2.** Streamflow [l/s] and water temperature [$^{\circ}$C] data of the 5 selected sampling sites during the two sampling campaigns.

| | February | | July | |
| --- | --- | --- | --- | --- |
| *Sampling sites* | *streamflow* [l/s] | *water temperature* [$^{\circ}$C] | *streamflow* [l/s] | *water temperature* [$^{\circ}$C] |
| WB2B | 85 | 3.4 | 610 | 12.6 |
| WB3A | 3366 | 3.6 | 13579 | 9.8 |
| WB3B | 3602 | 5.3 | 14529 | 13.1 |
| WB4B | 35742 | 7.7 | 28476 | 14.8 |
| WB5B | 36752 | 5.4 | 29281 | 13.7 |

measurements utilized in the present work is granted by the protocols used in the sampling campaign, the care in maintaining and shipping the samples, and the analytical methodologies used in the laboratory. For further details on the protocols followed in sampling, handling, shipping and analyzing the samples, we refer to the previous work of Mandaric et al. (2017).

Among the detected pharmaceuticals, the five with the highest concentrations in both sampling campaigns were selected for simulation. They are:

- Diclofenac: non-steroidal anti-inflammatory drug with antipyretic and analgesic actions;

- Ketoprofen: non-steroidal anti-inflammatory drug, analgesic and antipyretic;

- Clarithromycin: semisynthetic macrolide antibiotic;

- Sulfamethoxazole: sulfonamide bacteriostatic antibiotic. Its broad spectrum of activity has been limited by the development of resistance;

- Irbesartan: nonpeptide angiotensin II antagonist with antihypertensive activity.

Although detected only in February, diclofenac was included because it belongs to the watch list in the directive 2013/39/EU of the European Parliament. For additional information on these compounds we refer to the PubChem Compound database (PubChem).

The locations of WWTPs were obtained from the local authorities responsible for urban waste water treatment in the provinces of Trento (ADEP) and Bolzano (APPA, b). The geometry of the river network, including the distances of the WWTPs from the control sections, were obtained from the official river network shape file, which includes deviation of the natural river courses (EEA, 2017; ISPRA, 2015). All the spatial analyses were performed with QGIS (QGIS, 2018). Resident population and touristic presences were obtained from the census offices of the provinces of Trento (ISPAT) and Bolzano (ASTAT) at annual and monthly resolution, respectively. Only for the Noce sub-catchment the touristic presences were also available at the sampling days. Population was assigned to the WWTPs according to the served municipalities and resident population was assumed constant through the year. Daily streamflow time series (Q) were obtained from the hydrological offices of the

provinces of Trento (Ufficio-Dighe) and Bolzano (Ufficio-Idrografico). Monthly streamflow time series at the gauging stations were then computed by aggregating daily values. Finally, water temperatures (WT) at the streamflow gauging stations were provided by the Environmental Protection Agencies of the provinces of Trento (APPA, a) and Bolzano (APPA, b) at monthly resolution (Fig. 2). The parameters $\alpha_i$, $\beta_i$ and $D_i$ of Eq. (13) are obtained from the datasets of the Collaborating Centre for Drug Statistics Methodology of the World Health Organization (WHO) and the Italian Agency of Drug (AIFA).

## 4 Inference of the model parameters

Concentrations along the river network were predicted by an adaptation of Eq. (15) to include the effect of the S. Giustina and Mollaro Reservoirs. In the absence of information supporting a more accurate mixing model, we assumed that at a given time the concentration within the equivalent reservoir, simulating the effect of both reservoirs, is equal to the mean inflow load in the $\tau_s = 54$ previous days divided by the mean volume of the reservoirs (see Sect. 3, reduced as described below to take into account biological attenuation). Given that the load is parametrized by the amount of population and that the touristic presences are known only at the monthly scale, we assigned the average monthly load to each day of the month. According to Eq. (15) the load leaving the equivalent reservoir is computed by reducing the mass released from the upstream WWTPs by the factor $\exp\left[-k(\sum_{j=1}^{n_i} \tau_{i,j} + \overline{\tau}_s)\right]$, where $\overline{\tau}_s$ is the weighted reservoir residence time with respect to the population. This load is then divided by the average reservoir volume to obtain the mean outflow concentration. Here we utilized $\overline{\tau}_s$, instead of $\tau_s$, in order to take into account that at a given day the mass with age $\tau_a \in [0, \tau_s = 54\,days]$ contained into the reservoir depends on the number of persons within the catchment $\tau_a$ days before. In other words, the age distribution of solute particles contained into a volume of water sampled at the outlet of the lake is proportional to the temporal distribution of the population of the catchment in the $\tau_s$ days before. Notice that $\tau_{i,j}$ is in any case smaller than $1\,day$. This mixing model differs from the complete mixing one, which entails an exponential pdf, because the water (and the contaminant that it bears) entering at a given day is assumed to remain in the reservoir for 54 days while it mixes with the water entering up to 54 days before.

To comply with the Occam's razor principle (MacKay, 2003, ch. 28), suggesting parsimony in selecting model complexity and considering the very limited amount of concentration data available, the parameters in the Eq. (13) are assumed the same for all the WWTPs and since $\gamma$ is inferred, the abatement $f$ is assumed to be zero. This simplification is supported by the fact that the WWTPs of the two provinces are managed by the same agency by using similar technologies. The parameters space has been explored by Latin Hypercube sampling with the probability distribution assumed multi-log-normal with means and variances of $\gamma$ and $k$ provided in Table 3 and obtained from the pharmacological databases described in Sect. 3.2.

The inference of model's parameters was performed by using concentration measurements along the Noce river as observational variables. The unitary mass flux release $\gamma$ may change seasonally as an effect of variability in drugs consumption, due to changes in touristic fluxes, while the variability of $k$ depends on water temperature through the Arrhenius law (Eq. 18). Four candidate models with different numbers of parameters (i.e. $n_p$) were investigated:

- M1: a single value of $\gamma$ is considered through the year and decay $k$ is set to zero; this is a single parameter model ($n_p = 1$);

**Table 3.** Means ($\overline{\gamma}$, $\overline{k}$) and variances ($\sigma^2_{gamma}$, $\sigma^2_k$) of the log-normal distributions of the unitary mass fluxes ($\gamma$) and of the coefficients of decay ($\overline{k}$) for the 5 selected pharmaceuticals. The standard deviation of the associated normal distributions, with unitary means, was set equal to 3 in order to explore several order of magnitudes and, hence, all the plausible physical values.

| Pharmaceutical | $\overline{\gamma}$ [ng hab$^{-1}$d$^{-1}$] | $\sigma^2_\gamma$ [ng$^2$ hab$^{-2}$d$^{-2}$] | $\overline{k}$ [s$^{-1}$] | $\sigma^2_k$[s$^{-2}$] |
|---|---|---|---|---|
| **Diclofenac** | $2.09E+06$ | $1.37E+14$ | $1.87E-05$ | $1.23E+03$ |
| **Ketoprofen** | $1.19E+06$ | $7.82E+13$ | $2.00E-05$ | $1.31E+03$ |
| **Clarithromycin** | $8.98E+06$ | $5.90E+14$ | $3.25E-03$ | $2.13E+05$ |
| **Irbesartan** | $8.20E+05$ | $5.39E+13$ | $8.53E-05$ | $5.60E+03$ |
| **Sulfamethoxazole** | $4.21E+05$ | $2.77E+13$ | $4.42E-05$ | $2.90E+03$ |

- M2: two values of $\gamma$ are considered, one for the winter season and the other for the summer season, $k = 0$ as for M1; therefore $n_p = 2$;

- M3: a single value of $\gamma$ is considered, as in M1, while decay is assumed to vary with temperature according to the Arrhenius law (Eq. 18). This model requires $n_p = 3$ parameters, given that the Arrhenius law depends on $A$ and $E_A$;

- M4: $\gamma$ varying seasonally as in M2 and $k$ as in M3; therefore $n_p = 4$.

Since water temperature can be safely assumed constant in each sampling campaign, with the models M3 and M4 the inversion was performed by considering two values of $k$, one for the winter and one for the summer seasons, as unknowns instead of $A$ and $E_A$. Successively, the inferred values of k were used in Eq. (18), together with the water temperature to compute the parameters $A$ and $E_A$ of the Arrhenius law.

The inference was performed for each of the four selected models by searching the parameters hyperspace through Latin Hypercube Sampling (LHS) (McKay et al., 1979) with the objective to identify the set of parameters that minimizes the following weighted least-squares criterion (see e.g., Carrera and Neuman, 1986; McLaughlin and Townley, 1996; Tarantola, 2005):

$$L(a) = [z - \mathcal{F}(a)]^T C_v^{-1} [z - \mathcal{F}(a)] + [\overline{a} - a]^T C_a^{-1} [\overline{a} - a] \tag{19}$$

where $a$ is the vector of the unknown model parameters, $z$ is the vector containing the observational data, $\mathcal{F}$ is the output of the model at the measurement points (i.e. the Eq. 15 modified as discussed above), $C_v$ is the diagonal matrix of the error variances, $C_a$ is a diagonal matrix which epitomizes the effect of uncertainty associated to the prior information, and $\overline{a}$ is the vector of the prior estimates of the model's parameters (i.e. the means reported in Table 3). In addition, the superscript $T$ indicates the transpose of the vector. Under the commonly assumed hypothesis that the model's errors ($z - \mathcal{F}(a)$) and the residuals ($\overline{a} - a$)

are both normally distributed and independent, the minimum of the function (Eq. 19) coincides with the Maximum of the A-Posteriori (MAP) probability distribution (McLaughlin and Townley, 1996; Rubin, 2003; Castagna and Bellin, 2009).

LHS was performed by dividing the parameters axes in $N_L$ intervals of constant probability $1/N_L$, thereby resulting in a partition of the hypercube in $M = N_L^{n_p}$ cells. The upper boundary of the cell along the axis $a_j$, $j = 1, ...., n_p$ of the hypercube is obtained by inverting the cumulative log-normal probability distribution:

$$P(a_j) = \int_0^{a_j} \frac{1}{ln a_j \sqrt{2\pi\sigma_{y_j}^2}} \exp\left[ -\frac{\left(ln a_j - \bar{y}_j\right)^2}{2\pi\sigma_{y_j}^2} \right] da_j \tag{20}$$

at the following discrete values: $\{1/N_L, 2/N_L, ...., (N_L - 1)/N_L, 1\}$. In Eq. (20) the first two moments assume the following expressions:

$$\bar{y}_j = ln\left[ \frac{\bar{a}_j^2}{\sqrt{\bar{a}_j^2 + C_{a,jj}}} \right], \quad \text{and} \quad \sigma_{y_j}^2 = ln\left[ 1 + \frac{C_{a,jj}}{\bar{a}_j^2} \right] \tag{21}$$

where $C_{a,jj}$ is the $j-$th diagonal term of the matrix $C_a$. A sampling point is then generated randomly within each cell, thereby obtaining a total number of $M$ sampling points distributed within the hypercube. The sampling point that minimizes the function $L(a)$ given by Eq. (19) is recorded together with its value and the procedure is repeated $MC$ times, each time with a different random location within the cells. Inference is performed for each model with $M = 100$ and $MC = 10,000$. Parameters inference was repeated by using the Nash-Sutcliffe Efficiency Index (NSE; Nash and Sutcliffe, 1970) as the objective function (this time by maximizing $NS$), obtaining optimal sets of parameters close to those identified by MAP.

The choice among the four models, each one with the optimal parameter set, has been performed by using the Akaike's information criterion (AIC; Akaike, 1974), which penalizes models with more parameters (Akaike, 1987):

$$AIC = p ln\left( \frac{[z - \mathcal{F}(a)]^T [z - \mathcal{F}(a)]}{p} \right) + 2 n_p \tag{22}$$

where $p$ is the number of experimental data points and $n_p$ is the number of parameters.

The highest AIC values are obtained with the model M4 (217, on average for the 5 pharmaceuticals), which is then discarded. For this model the better fit, granted by the larger number of parameters, is not enough to justify higher model complexity, according to the Akaike criterion. Also the less complex model (i.e. M1) was discarded because of the poorer fit with the observations, despite the relatively low Akaike number (AIC=206), with respect to models M2 and M3, both with a AIC approximately equal to 200. Given the importance of including seasonality in modeling bio-geochemical processes, the model M3 was preferred to the model M2, despite less parsimonious in term of number of parameters (three instead of two). The comparison between observations and modeling results by the model M3 for the 5 selected compounds are shown in the Fig. 3 with the optimal set of parameters shown in Table 4. Despite the lower number of parameters, concentrations of diclofenac along the Noce river are reproduced very well by a simplified version of the model M3 with two parameters. The likelihood function is two orders of magnitude smaller than for the other compounds ($L = 5.13$, see Table 4) and predicted concentrations

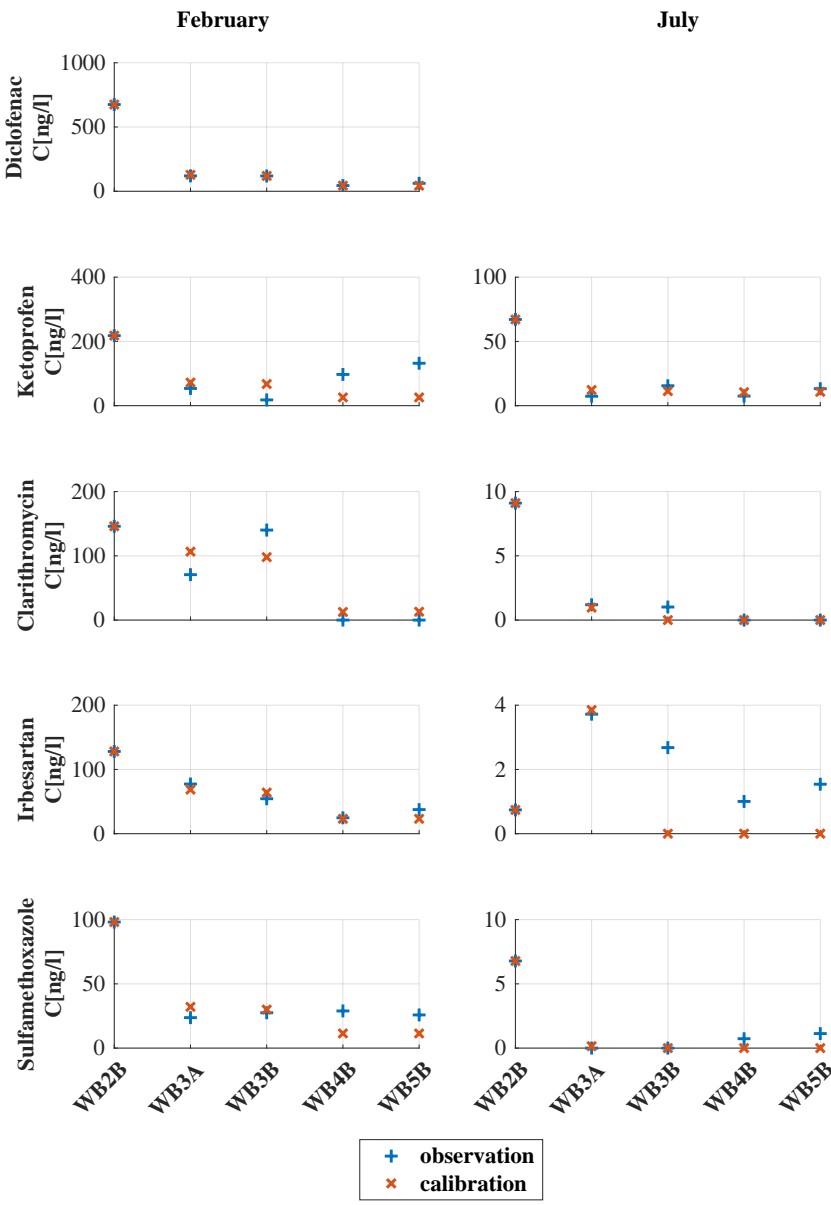

**Figure 3.** Concentrations of 5 selected pharmaceuticals for both winter and summer campaigns. Red crosses represent modeling results obtained with the model M3, whereas blue plusses represent measured concentrations at the 5 selected locations along the Noce river. Diclofenac was detected only during the winter campaign.

**Table 4.** Model M3 theoretical load coefficients ($\gamma$), coefficients of decay ($k_{february}$ and $k_{july}$) and $L$-values for the 5 pharmaceuticals.

| *Pharmaceutical* | $\gamma\ [\mathrm{ng\,hab^{-1}d^{-1}}]$ | $k_{february}\ [\mathrm{s^{-1}}]$ | $k_{july}\ [\mathrm{s^{-1}}]$ | $L\ [-]$ |
|---|---|---|---|---|
| **Diclofenac** | $2.07E+06$ | $1.33E-10$ | $-$ | $5.13$ |
| **Ketoprofen** | $1.24E+06$ | $4.23E-10$ | $1.27E-06$ | $62.93$ |
| **Clarithromycin** | $1.93E+06$ | $1.65E-06$ | $5.31E-03$ | $34.20$ |
| **Irbesartan** | $1.22E+06$ | $4.51E-08$ | $1.70E-03$ | $75.95$ |
| **Sulfamethoxazole** | $5.53E+05$ | $1.39E-09$ | $6.59E-03$ | $332.47$ |

are almost indistinguishable from the observed ones (Fig. 3). The parameters of M3 in this case are 2, instead of 3, because no diclofenac was detected in summer and therefore only the winter $k$ value was inferred from the data. The concentration of the other compounds are well reproduced by the model M3 in both seasons, except irbesartan in summer at the sampling locations WB3B, WB4B and WB5B and to some extent also ketoprofen in winter, particularly at the sampling locations WB4B and WB5B. The pharmaceutical with the highest L-value (i.e. the worst match with observations) is sulfamethoxazole with $L = 332.47$ (Table 4). Overall, the model M3 was able to capture the observed concentrations of PPCPs along the Noce river in both winter and summer campaigns. As expected (see Table 4), the inferred values of the decay rate $k$ are lower in winter than in summer for all the compounds: this is due to the temperature dependence of biological processes causing degradation.

Figure 4 shows the likelihood function $L$ given by Eq. (19) as a function of the model parameters for the model M3. In particular, $L$ is represented in a two dimensional space as a function of $\gamma$ and $k$, the latter being different in the two sampling campaigns (see the two columns of Fig. 4). Notice that the model allows a clear identifiability of the parameters for all the compounds, as shown by the relatively small dark blue areas corresponding to low $L$ values, located at large distance from the boundaries of the parameters space.

All the computations for the inference of the model parameters and the following application of the model illustrated in Sect. 5 have been performed by coding the model with MATLAB (MATLAB, 2017).

## 5 Application at catchment scale

As an illustrative example, we applied our modeling framework to the whole Adige river with the objective of evaluating seasonal variations of PPCPs concentrations at a few relevant locations. The simulations were limited to the year 2015 for illustration purposes, but they can be extended over longer periods, including future projections, if hydrological and population data, both recorded or modeled are available. The model, in particular, allows considering the interplay between hydrological and population variability, the latter due to touristic fluxes. Simulations were performed by using the model M3 with the optimal parameters shown in Table 4 and the S. Giustina and Mollaro reservoirs modeled as described in Sect. 4. The other reservoirs of the Adige catchment are either not intercepting release points or have a small volume and, therefore, were not

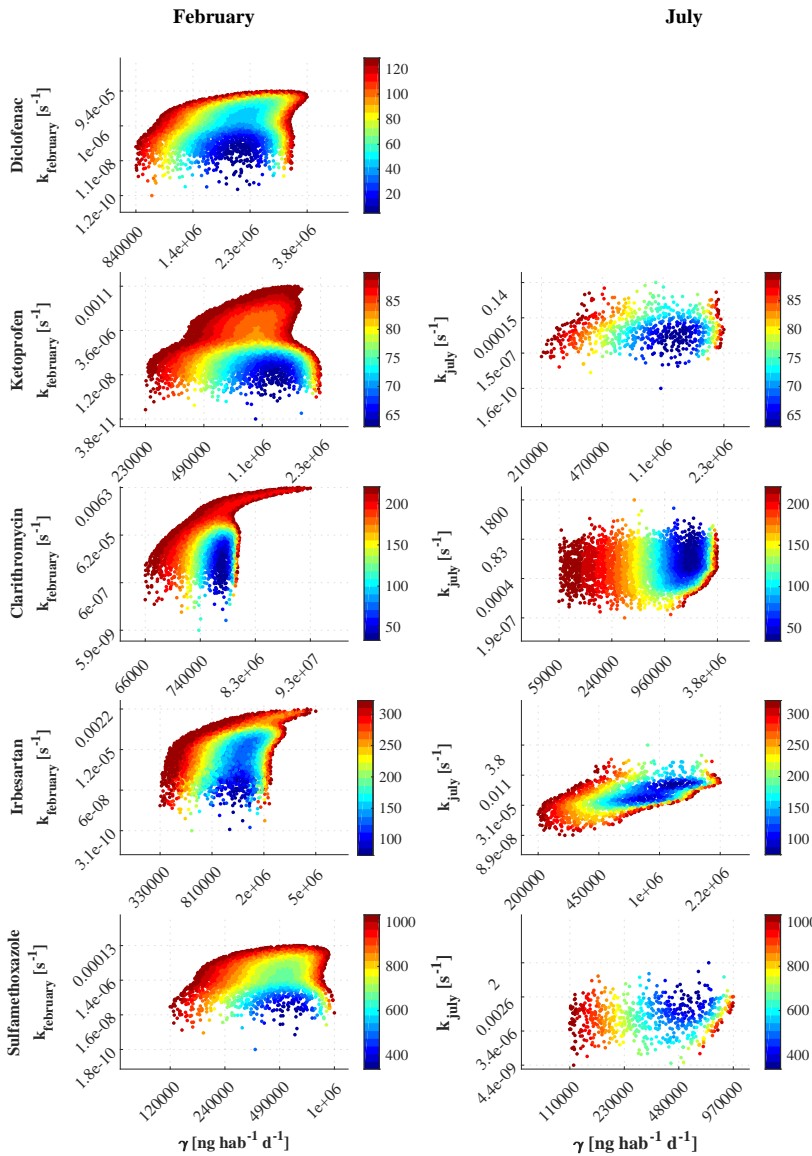

**Figure 4.** Objective function $L$, given by Eq. (19), as a function of the model's parameters for the 5 selected compounds. The left and right columns refer to winter and summer champaign, respectively. Each dot at the position sampled by the latin hypercube technique assumes the color corresponding to the scale for $L$ shown on the right of each panel. For clarity, pairs of $k$ and $\gamma$ resulting in values of $L$ larger than the 0.1 quantile are not shown.

included. Only WWTPs with maximum served population above 10,000 persons-equivalent were selected as release points for the simulations, and communities served with systems of smaller size were aggregated to the closest release point. Altogether, 26 WWTPs release points were included into the model, obtaining a good spatial coverage (see Fig. 2). The monthly average number of persons actually served was obtained by aggregating the municipalities and the census data (including the touristic presences) to each release point .

It should be acknowledged that model's parameters are affected by uncertainty, which is expected to be large due to the limited number of data available for inference. For this reason the results of the simulations discussed here should be considered as a preliminary exploration providing uncertain estimates of concentrations at the sampling points. This limitation is due to the lack of proper data and cannot be, by any means, attributed to limitations in the structure of the model.

The decay rates ($k$), evaluated at the monthly scale, were obtained by means of the Arrhenius kinetics parameters ($A$ and $E_A$), considered constant for each one of the 5 pharmaceuticals and calculated from Eq. (18), given the two $k$ values obtained by inversion of the observational data (see Table 5). For diclofenac the summer value of $k$ was inherited by ketoprofen, which belongs to the same pharmacological class. Notice that $k$ varies by orders of magnitude (i.e. in the range $10^{-10} - 10^{-3}$ [s$^{-1}$]), between winter and summer, due to the seasonal fluctuations of water temperature. In winter the decay rate is rather small for all compounds, suggesting dilution as the main attenuation mechanism which, on the other hand, in winter is at its minimum due to low streamflow. In summer the decay rate is significantly higher, resulting in a concurrent effect of biological decay and dilution, which is higher than in winter due to snowmelting, for all the compounds. In terms of half-life time, July is the month with the fastest decays (half life of 95 h for diclofenac and ketoprofen, 2.6 min for clarithromycin, 7 min for irbesartan and 22 s for sulfamethoxazole), whereas December and January are the months with the slowest decays (half life of 43 y for diclofenac, 15.4 y for ketoprofen, 6.4 d for clarithromycin, 3.8 y for irbesartan and 4.5 y for sulfamethoxazole). Notice that in winter none of the 5 compounds analyzed in the present study can be considered as biologically decaying, since their half life is significantly larger than the residence time.

Figure 5 shows the annual average of the simulated concentrations for the 5 compounds (panels from (a) to (e)) along the Adige main stem and its tributaries. The highest mean concentrations were obtained for diclofenac (panel a), particularly in the middle course of the main stem and in the Eastern portions of the basin. The relatively high concentrations in the upper Rienza and Avisio rivers (see Fig. 2 for the location of the Adige's tributaries) is a consequence of the combined effect of low dilution and high PPCPs load due to the touristic presences. Intermediate values are observed in the Noce middle stem and in the southernmost portion of the Adige river. Also ketoprofen (panel b) shows concentrations higher than $100$ ng/l, in particular downstream of the WWTP of Bolzano, labelled as 14 in Fig. 2 (see Table B1 in the Appendix B). For all the compounds, concentrations are relatively high in the headwaters of the Rienza river (i.e. upper Gadera catchment), where the touristic presences are high in winter. In general, concentrations of all pharmaceuticals show a remarkable spatial variability with values ranging from 0 to $200$ ng/l with maximum in the central and North-Eastern portion of the basin. This entails that local pharmaceutical consumption affects remarkably the detected concentrations in rivers and, in some cases, overwhelms natural dilution which varies linearly with water discharge. Notice that in the lower part of the Adige main stem, after the confluence

**Table 5.** Monthly decay rates ($k$ [s$^{-1}$]) of the 5 selected compounds, computed by using the Arrhenius law (Eq. 18) with the values of $A$ [s$^{-1}$] and $E_A$ [kJmol$^{-1}$] , obtained by inverting Eq. (18) with reference to the decay rate inferred from the observational data of February and July 2015 (first two rows).

| | Diclofenac | Ketoprofen | Clarithromycin | Irbesartan | Sulfamethoxazole |
|---|---|---|---|---|---|
| $A$ [s$^{-1}$] | $9.32E+72$ | $2.17E+73$ | $2.46E+79$ | $5.60E+117$ | $1.15E+152$ |
| $E_A$ [kJmol$^{-1}$] | $4.36E+02$ | $4.36E+02$ | $4.51E+02$ | $6.65E+02$ | $8.51E+02$ |
| $k\,(January)$ [s$^{-1}$] | $2.20E-10$ | $5.16E-10$ | $1.28E-06$ | $7.90E-09$ | $8.05E-10$ |
| $k\,(February)$ [s$^{-1}$] | $2.70E-10^*$ | $6.34E-10^*$ | $1.58E-06^*$ | $1.08E-08^*$ | $1.20E-09^*$ |
| $k\,(March)$ [s$^{-1}$] | $3.33E-09$ | $7.82E-09$ | $2.11E-05$ | $4.96E-07$ | $1.61E-07$ |
| $k\,(April)$ [s$^{-1}$] | $1.18E-08$ | $2.77E-08$ | $7.79E-05$ | $3.40E-06$ | $1.90E-06$ |
| $k\,(May)$ [s$^{-1}$] | $4.38E-08$ | $1.03E-07$ | $3.02E-04$ | $2.51E-05$ | $2.45E-05$ |
| $k\,(June)$ [s$^{-1}$] | $8.94E-08$ | $2.10E-07$ | $6.31E-04$ | $7.45E-05$ | $9.88E-05$ |
| $k\,(July)$ [s$^{-1}$] | $7.37E-07$ | $1.73E-06^*$ | $5.56E-03^*$ | $1.85E-03^*$ | $6.04E-03^*$ |
| $k\,(August)$ [s$^{-1}$] | $1.82E-07$ | $4.26E-07$ | $1.31E-03$ | $2.19E-04$ | $3.93E-04$ |
| $k\,(September)$ [s$^{-1}$] | $3.37E-08$ | $7.92E-08$ | $2.31E-04$ | $1.69E-05$ | $1.48E-05$ |
| $k\,(October)$ [s$^{-1}$] | $6.94E-09$ | $1.63E-08$ | $4.51E-05$ | $1.52E-06$ | $6.75E-07$ |
| $k\,(November)$ [s$^{-1}$] | $3.81E-09$ | $8.94E-09$ | $2.43E-05$ | $6.08E-07$ | $2.09E-07$ |
| $k\,(December)$ [s$^{-1}$] | $2.20E-10$ | $5.16E-10$ | $1.28E-06$ | $7.90E-09$ | $8.05E-10$ |

$^*$ inferred $k$ values.

with the Noce and Avisio tributaries, concentrations decrease for all the compounds. This is due to both the attenuating effect of dilution at the nodes and mixing within both S. Giustina and Mollaro reservoirs, in the middle course of the Noce.

During the two sampling campaigns, samples were collected and concentrations evaluated at the sites WB6, just upstream of the city of Trento and the confluence of both Noce and Avisio, and at four locations labelled WB7 A, B, C and D, downstream of Trento (see Mandaric et al. (2017) for locations). This additional information cannot be used for a formal validation because representative of the sampling day, while simulations are conducted at the monthly scale. Indeed, along the main stem of the Adige river, census data are available only at monthly scale for the touristic fluxes and at the annual scale for the resident population. While one can safely assume that resident population changes little within a year, touristic fluxes show significant variations at the weekly and even shorter time scales. Monthly concentrations produced by the model at selected sections are discussed below keeping in mind this limitation.

Figure 6 shows the monthly average of flux concentrations at the following 4 selected control sections (see Fig. 2): Soraga in the upper Avisio river; Vermiglio in the Vermigliana creek (headwaters of the Noce river); Ponte Adige along the Adige river before the confluence with the Isarco river (draining the Northwestern portion of the Adige catchment) and Bolzano in the Isarco river at the confluence with the Adige river. These control sections were selected because official gauging stations with a significant drainage basin. Diclofenac is present at all gauging stations with concentrations up to 300 ng/l at Isarco

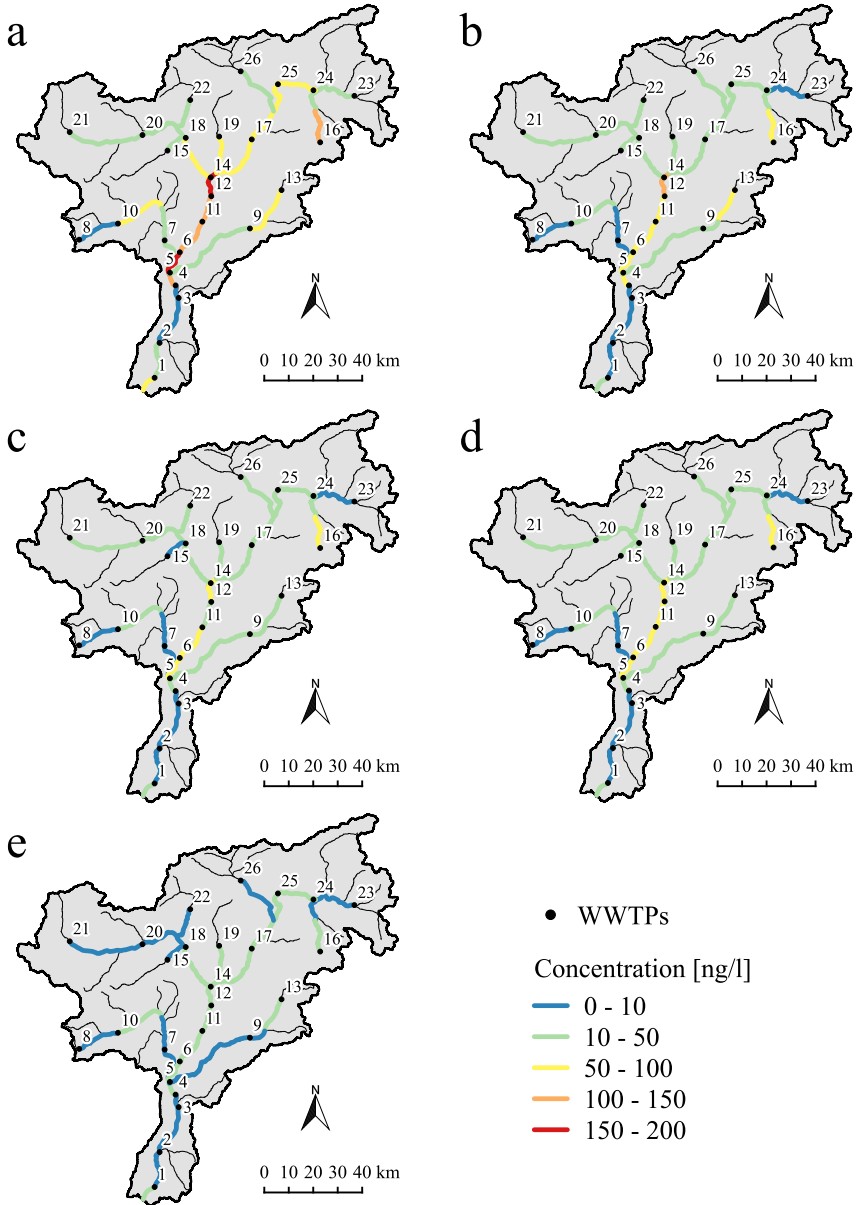

**Figure 5.** Annual mean concentrations of (a) diclofenac, (b) ketoprofen (c) clarithromycin (d) irbesartan (e) sulfamethoxazole in the Adige catchment for the year 2015. The position of WWTPs along the river network are marked with a black bullet and numbered progressively from the Southern control section in the main stem of the Adige river to the headwaters (see Table B1 in the Appendix B). Color's scale is from blue (low concentration) to red (high concentration).

at Bolzano. For comparison see also the annual mean values shown in Fig. 5. On the other hand, sulfamethoxazole shows the lowest concentrations, due to the modest input loads (see Table 4). The temporal pattern of diclofenac and ketoprofen

is characterized by two peaks, one between February and March and the other in August, following the seasonal pattern of touristic fluxes. The other pharmaceuticals are without the summer peak, but show a slight increase in autumn before the winter peak. Besides touristic presences, these patterns are shaped by the interplay between dilution and decay, which are both higher in summer than in winter due to snow melting and higher temperatures, respectively. The importance of streamflow seasonality is clearly evident in the low concentrations observed between April and June when snow melting and therefore streamflow are at their maximum. Decay, instead, is more effective in reducing the summer concentrations of clarithromycin, irbesartan and sulfamethoxazole rather than that of the anti-inflammatories, according to their higher decay coefficients (see Table 4 and Table 5). At Vermiglio the second peak in August is less pronounced with respect to the other control sections. This attenuation is justified by lower touristic fluxes with respect to winter (i.e. 2,824 persons per day on average served by the WWTP at Passo del Tonale in August 2015 against 4,166 in February of the same year) and by streamflow contribution from summer melting of Presanella and Presena glaciers which maintains high streamflow also after the end of the snow melting season (see e.g., Chiogna et al., 2016, and Table 2). The highest peak at Soraga is observed in August, and this is in agreement with the higher touristic presences in summer with respect to winter (about 25,000 and 19,000 persons per day served by the upstream WWTP of Pozza di Fassa in August and February 2015, respectively), while the contribution from the Marmolada glacier is diverted outside the basin through the Fedaia reservoir (see e.g., PAT, 2012), thereby reducing dilution. Conversely, at Ponte Adige the winter peak is higher than the summer one, showing a complex interplay between variability of streamflow and touristic presences. Also at Bolzano the simulated concentrations are higher in winter than in summer. In addition, here the modeled concentrations are higher than in the other control sections (see also Fig. 5).

These simulations showed that concentrations of PPCPs changed through the seasons depending on both population dynamics and hydrological characteristics of the year. This behavior cannot be identified when a single emission and a representative water discharge are adopted, as commonly done in applications.

## 6  Discussion and Conclusions

In the present paper we proposed a simplified, yet realistic, transport model of pharmaceuticals and personal care products in a river network. The model takes into account time variability of both hydrological fluxes and emissions from the waste water treatment plants, and other sources, at daily or larger scales, thereby overcoming the main limitations of the existing approaches, which assume both processes as time invariant. Emissions are computed by considering available data on consumption of pharmaceuticals and personal care products and population, including touristic fluxes, which are becoming more and more important since touristic activities expanded tremendously in the last decades. Attenuation processes are dilution and biogeochemical decay, the former included by considering streamflow variable at the proper scale and the latter by a first-order irreversible decay reaction. The effect of lakes, or reservoirs, is included in our modeling approach by adding their residence time pdfs, which shapes depend on the mixing characteristics of the reservoirs, to the convolution chain.

The model was applied to the Adige river basin, North-East of Italy, by considering a selection of five pharmaceuticals, which presence was detected in two sampling campaigns conducted in February and July 2015, and belonging to the groups

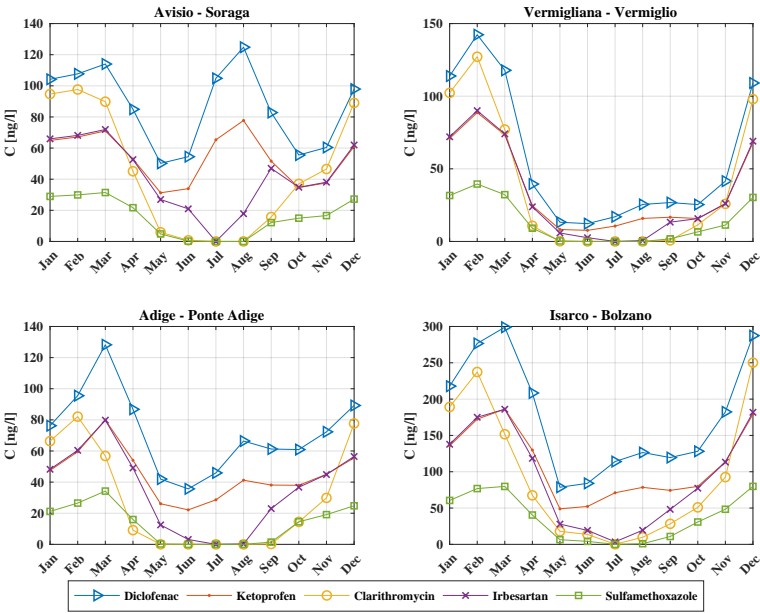

**Figure 6.** Monthly flux concentrations of 5 PPCPs (diclofenac, ketoprofen, clarithromycin, irbesartan and sulfamethoxazole) at the control sections of Soraga in the Avisio river, Vermiglio in the Vermigliana creek, Ponte Adige in the Adige river and Bolzano in the Isarco river. Concentrations are expressed in $ng/l$.

of analgesic/anti-inflammatory, antibiotic and antihypertensive. Four parametrizations of the model, corresponding to different hypotheses on the variability of the per-capita emission rate (i.e. $\gamma$) and the decay rate (i.e. $k$), have been considered and the corresponding parameters were obtained by inversion of the observational data collected in the two sampling campaigns. The best performing parametrization was identified according to the Akaike information criterion. The selected parametrization of the model includes a constant in time $\gamma$, such that variability of the emissions follows changes in the population, which is high in the Alpine area due to important touristic fluxes in winter and summer seasons, and a decay rate $k$ varying as a function of water temperature through the Arrhenius law. This three-parameter model has been applied at monthly time scale to the whole Adige catchment for the year 2015. The inferred monthly concentrations at 5 control sections not used for calibration are compatible with those measured during the sampling campaigns, revealing the capability of the model to reproduce spatial and temporal patterns of concentration. However, the lack of data at the proper time scale did not allow to perform a formal verification of the model.

Monthly flux concentrations at four relevant gauging stations showed a significant seasonal variability as it was expected considering the large fluctuations of touristic presences and the strong seasonality of streamflow. In general, decay processes, as epitomized by the decay rate $k$, are less important than dilution due to both streamflow variability and the contribution from sub-catchments slightly or not impacted by pharmaceutical releases. In winter, when dilution is low, the decay rate is also at its minimum because of the low water temperature. On the other hand, in summer the relevance of a high decay rate due to the

high temperature is diminished by the overwhelming effect of dilution. Among the five selected pharmaceuticals, diclofenac and ketoprofen are those less affected by decay, while clarithromycin and sulfamethoxazole are the compounds subject to the highest decay. Overall, the proposed modeling approach shows seasonal and spatial patterns of solute concentrations in the stream water, which cannot be detected with existing approaches inherently time invariant. Touristic fluxes and streamflow variability are the driving factors of these patterns, and should be carefully estimated in applications. Indeed, the effects of streamflow and touristic fluctuations in the concentration patterns are intertwined, to an extent that depends on the particular compound and local conditions, and worth to be further analyzed to obtain reliable estimates of their impact on the freshwater ecosystem. Finally, our modeling framework is structured in a way that allows its use in combination with hydro-climatological models to elaborate future scenarios.

*Code and data availability.* The MATLAB code of the model and the data obtained from the sampling campaigns are available upon request. All the other data needed for the reproducibility of the results are freely available and the sources are reported in Sect. 3.2.

## Appendix A: Scaling coefficients of the stream velocity

Dodov and Foufoula-Georgiou (2004) proposed the following scaling laws for the coefficients $\Phi$ and $\Psi$ of the geometry hydraulic expression (Eq. 3):

$$\Phi(A) = \exp\left[-\left(\alpha_{C_A} + \beta_{C_A} \ln(A)\right) + \left(\alpha_Q + \beta_Q \ln(A)\right)\Psi_{C_A}\right] \tag{A1}$$

with

$$\Psi_{C_A}(A) = \left[\frac{\gamma_{C_A} + \delta_{C_A}\ln(A)}{\gamma_Q + \delta_Q\ln(A)}\right]^{1/2} \tag{A2}$$

and finally the exponent $\Psi$ is given by:

$$\Psi(A) = 1 - \Psi_{C_A}(A) \tag{A3}$$

In all these expressions $A$ [km$^2$] is the contributing area and the other coefficients are reproduced in the Table A1 (see also Table 3 of the paper by Dodov and Foufoula-Georgiou (2004)).

**Table A1.** Parameters of the scaling coefficients by Dodov and Foufoula-Georgiou (2004).

| Hydraulic geometry factors | $\alpha$ | $\beta$ | $\gamma$ | $\delta$ |
|---|---|---|---|---|
| $C_A$ | $-3.1802$ | $0.6124$ | $0.8404$ | $0.1130$ |
| $Q$ | $-5.5428$ | $0.7992$ | $2.6134$ | $0.0012$ |

## Appendix B:  Location of the Waste Water Treatment Plants of the Adige river

The Table B1 shows the identification number used in the map of Fig. 2 to locate the WWTPs with maximum capacity larger than 10,000 person-equivalent installed in the Adige catchment.

**Table B1.** List of the WWTPs with maximum capacity of more than 10,000 person-equivalent considered in the application at catchment scale. The assigned identification numbers (ID) are sorted by latitude (from the southernmost to the northernmost plant) and the coordinates are expressed in [m] (UTM WGS 84).

| ID | Name | Easting [m] | Northing [m] |
|----|------|-------------|--------------|
| 1 | Ala | 654586 | 5067957 |
| 2 | Rovereto | 656654 | 5082229 |
| 3 | Trento Sud | 664445 | 5100590 |
| 4 | Trento Nord | 663193 | 5105707 |
| 5 | Lavis | 660857 | 5110902 |
| 6 | Mezzocorona | 664890 | 5119257 |
| 7 | Campodenno | 658702 | 5124097 |
| 8 | Passo Tonale | 623766 | 5124390 |
| 9 | Tesero | 693548 | 5128973 |
| 10 | Mezzana | 639633 | 5130966 |
| 11 | Termeno | 674101 | 5131753 |
| 12 | Bronzolo | 677758 | 5142131 |
| 13 | Pozza di Fassa | 706482 | 5144652 |
| 14 | Bolzano | 677544 | 5149815 |
| 15 | San Pancrazio | 659866 | 5160771 |
| 16 | Sompunt | 722274 | 5164161 |
| 17 | Bassa Valle Isarco | 694302 | 5165355 |
| 18 | Merano | 667329 | 5166009 |
| 19 | Sarentino | 680981 | 5166517 |
| 20 | Media Val Venosta | 649646 | 5167136 |
| 21 | Alta Val Venosta | 619907 | 5168345 |
| 22 | Passiria | 669174 | 5181457 |
| 23 | Wasserfeld | 736247 | 5183152 |
| 24 | Tobl | 719504 | 5185433 |
| 25 | Bassa Pusteria | 704999 | 5187904 |
| 26 | Wipptal | 689775 | 5193172 |

*Competing interests.* The authors declare that they have no conflict of interest.

*Acknowledgements.* This research received financial support by the European Union under the 7th Framework Programme (Grant agreement no. 603629-ENV-2013- 6.2.1-Globaqua) and by Fondazione CARITRO (Rif. int. 2018.0288 - Bando 2018 per progetti di ricerca svolti da giovani ricercatori post-doc). We thank the Environmental Protection Agencies and Hydrological and Meteorological Offices of the Autonomous Provinces of Trento and Bolzano for providing the hydrological data and the ISPAT Office of the Province of Trento for providing the data on touristic fluxes. We heartfelt thanks the graduate student Deborah Bettoni for her precious collaboration in data collection and organization.

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
