# Peer review of "A parsimonious transport model of emerging contaminants at the river network scale"

_Hydrology and Earth System Sciences, 2018_

## Referee Comment (RC1) · Anonymous Referee #1 · 17 Jul 2018

General comments:

This manusript presents a simple approach to estimate concentrations of pharmaceuticals and personal care products (PPCP) in river networks through approximations of waste water treatment plant (WWTP) inflow and in-channel dilution. WWTP inflow is related to capita connected and thus an extrapolation to ungauged sites is theoretically possible, although the model requires local calibration to measured concentrations. While the general novelty of this approach is understood, the present manuscript lacks an adequate description of existing knowledge and a model check with data that was not used for calibration. Since independent data obviously exists – i.e. was published

in a preceding paper - such a model check is possible and would strongly contribute to the value of this research. Moreover, a large Lake of 180 mio m3 exists in the middle of the examined river network but is neither shown nor mentioned or adequately discussed. Such a lake will certainly affect model results and probably also violate underlying model assumptions. Only if all concerns are adequately addressed, publication in HESS is warranted. I detail these concerns down below.

Specific comments:

1. Existing knowledge: Concentrations of PPCPs

The paper lacks a description of existing studies on the occurrence and spatio-temporal dynamics of PPCPs in European rivers. In recent years several studies have been published in various river systems. This is important, because the results of the present studies should be compared to other regions to prove that both measured and simulated concentrations are realistic. I propose a table in the introduction where the five simulated compounds are selected and occurring concentrations are given for different rivers. Those should be used to evaluate model quality in the discussion. By the way: it is not true that PPCPs are not measured on a regular basis by environmental agencies: At least in the Rhine River measurement stations exist for this purpose (e.g. in Weil, check at: http://www.aue.bs.ch/umweltanalytik/rheinueberwachungsstation-weil-am-rhein.html

2. Existing knowledge: Model approaches

The literature overview at the end concentrates on two, relatively similar GIS-based model approaches, GREAT-ER and PhATE, that are both more than ten years old. More recent approaches are missing, e.g. the study of Osorio et al (2012) who applied a simple plug-flow model to simulate pharmacologically active compounds in the Ebro river taking into account different dilution by varying flow conditions and an overall decay constant. They also determined decay constants for ibuprofen, furosemide, enrofloxacin, enalapril, acetaminophen, diclofenac and ketoprofen. So the authors should

state why they developed a new model and what are the differences and benefits of their approach. Finally they should also compare their calibrated decay constants to those by (Osorio et al. 2012).

3. Missing information about model boundary conditions

How was the weather (rainfall, air/water temperature, etc.) during the two sampling periods? Since water temperature and discharge are two main parameters that influence the model results, this information is crucial to evaluate the model results. If figure 3 of Mandaric et al. (2017) is considered, the runoff difference between the sampling campaigns of February and July is rather small for the two downstream sites WB4B and WB5B. Here absolute numbers should be presented in a table, in addition to weather conditions and air/water temperatures.

4. Missing information about Lake di Santa Giustina

A more accurate map of the study catchment is required. Unlike in the preceding paper, an existing large lake between upstream (WB3A and WB3B) and downstream (WB4B and WB5B) sites is not shown in the catchment map and even not mentioned in the paper. Why? This lake of 180 million m3 volume will largely increase residence times and mixing and certainly affect model results. This could also explain differing model results between the upstream and downstream stations. Moreover, such a large lake has impacts on various simplifying model assumptions, as discussed below. Here an extensive discussion is needed including a quantitative estimate on the solute residence times in the lake (at different discharge conditions).

5. Model assumptions

Limitations caused by the following model assumption need adequate discussion:

a.) the model only accounts for point sources. How large is the error by diffuse sources, e.g. sewer overflows during large rainfall events, manure use, etc. b.) the entire concept is based on the hypothesis that the scale of interest is larger than the residence

time. Does this hold, especially regarding the big lake? c) can velocity be assumed to be spatially uniform within the channels (p 4)? Generally it seems more plausible that flow velocities are higher in steep headwaters and again, what about the lake? d) also the assumption that local dispersion is overwhelmed by geomorphological dispersion is problematic if one thinks about a large lake. e) how realistic is the assumption that every WWTP is equally efficient, i.e. has the same decay factor (p. 10)? From my experience there are large differences depending on the size of the WWTP, on its age and used techniques. Snip et al. (2014) exemplified the effect of different operation conditions on WWTP efficiency regarding PPCP removal. f) was the water temperature really constant throughout the network and throughout the three-day sampling campaign? Here again the balancing effect of the lake might play a role. Real measured data needs to be presented here (see 3. above).

6. Model check

Apart from the missing incorporation of the big lake, this is the main criticism of the presented approach: a missing model check with independent data. In the chapter 4 the model is extrapolated to the entire Adige river catchment and the spatial (Figure 5) and temporal (Figure 6) dynamics of selected components are shown. But how realistic are these results? Here the authors miss the chance of an independent model check. In their preceding paper Mandaric et al. (2017) presented additional data of the Adige catchment upstream the inflow of the Noce and downstream the town of Trento. This data could be used for a real model check and compared to simulated concentration WITHOUT re-calibrating the model.

7. Model description

A large part of the present manuscript consists of a presentation of the applied model approach, model selection and calibration strategy. The model description is lengthy and should be condensed. The section 2 "model" should be included into the section "material and methods". Here the model development itself should be described in a

first subsection followed by parameterization and calibration strategies. Most of the equations should be moved into the annex, and figure 1 be omitted.

8. Code and data availability

The sampled data should be provided as a table in the annex (upon publication the data is already published twice). Also the model code could be provided as an executable file.

Technical corrections:

The entire manuscript needs a thorough language check by a native speaker. Internet-Links should be omitted from the main text.

References:

Mandaric, L. et al. (2017): Contamination sources and distribution patterns of pharmaceuticals and personal care products in Alpine rivers strongly affected by tourism, Science of The Total Environment, 590, 484–494.

Osorio V. et al. (2012): Occurrence and modeling of pharmaceuticals on a sewage-impacted Mediterranean river and their dynamics under different hydrological conditions, Science of the Total Environment 440, 3–13.

Snip et al., (2014): Modelling the occurrence, transport and fate of pharmaceuticals in wastewater systems, Environmental Modelling & Software 62, 1-16.

---

## Referee Comment (RC2) · Anonymous Referee #2 · 29 Jul 2018

The manuscript describes a model of transport of contaminants in surface water. Specifically, five pharmaceuticals (diclofenac, ketoprofen, clarithromycin, sulfamethoxazole and Irbesartan) are modelled in a case study application. The proposed framework is original and the main innovative contribution is related to the stationarity hypothesis and to the number of parameters.

The manuscript is well written and pleasant to read. The model is appropriately described and the case study is rich of information. Results are promising and the proposed model represents an interesting amelioration compared to the previous ones present in literature.

[Figure]

I have only a minor suggestion, that is, to improve the figure 3 (first line: diclofenac) since apparently it is not clear that there are also blue points under the red ones.

---

## Author Comment (AC1) · 12 Sep 2018

We thank the Reviewer for his/her appreciation of our work. In case of a positive response by the Editor, we will improve figure 3 by substituting blue points with other symbols.

---

## Author Response (AR1)

Dear Editor and Reviewers,

The throughout reviews that we received contributed to improve the manuscript in several parts. Given this additional effort, we decided to involve Stefano Mallucci, who contributed making new simulations and participated in the revision of the manuscript. Given his contribution, both the authors of the original manuscript (Bellin and myself) feel that he deserves to be included as author. Therefore, according to the Editor and the Copernicus team, we ask for updating the list of the authors in the following order: Elena Diamantini, Stefano Mallucci and Alberto Bellin. We would be grateful for your understanding.

With sincere regards,

Trento, 25 October 2018

Elena Diamantini (corresponding author)

University of Trento

Department of Civil, Environmental and Mechanical

Engineering

Via Mesiano 77, 38123 Trento, Italy

Phone +39-340-4118436

elena.diamantini@gmail.com

NOTE: *Comments from the Reviewers are in BLUE and ITALIC.*

**Reply to Referee #1**

*COMMENTS FROM Referee #1*

*General comments:*

*This manusript presents a simple approach to estimate concentrations of pharmaceuticals and personal care products (PPCP) in river networks through approximations of waste water treatment plant (WWTP) inflow and in-channel dilution. WWTP inflow is related to capita connected and thus an extrapolation to ungauged sites is theoretically possible, although the model requires local calibration to measured concentrations.*

*While the general novelty of this approach is understood, the present manuscript lacks an adequate description of existing knowledge and a model check with data that was not used for calibration. Since independent data obviously exists – i.e. was published in a preceding paper - such a model check is possible and would strongly contribute to the value of this research. Moreover, a large Lake of 180 mio m3 exists in the middle of the examined river network but is neither shown nor mentioned or adequately discussed. Such a lake will certainly affect model results and probably also violate underlying model assumptions. Only if all concerns are adequately addressed, publication in HESS is warranted. I detail these concerns down below.*

**Reply**

We would like to thank the reviewer for evaluating our manuscript and for his/her valuable comments. Below we reply point by point to the reviewer's comments and illustrate the changes implemented in the revised manuscript.

To summarize, we address the three main concerns of the reviewer as follow:

**concern # 1**) "*the present manuscript lacks an adequate description of existing knowledge*": we partially agree with this statement. We expand the discussion on the existing models, including also the one suggested by the reviewer. However, we do not agree that the existing knowledge was not adequately described in the original version of the manuscript as explained below in the answer to the specific comment by the reviewer;

**concern # 2**) "*effect of the lake*": we deal with this important point raised by the reviewer as explained below by expanding our model's description to include the effect of reservoirs and lakes to demonstrate its flexibility in dealing with impacted catchments;

**concern # 3**) "*a model check with data that was not used for calibration*": Unfortunately we cannot make the validation suggested by the reviewer. The

additional data he/she mentions are very few and most importantly we cannot perform for these points simulations at the daily scale because only monthly touristic presences are available for the Adige catchment, except in the Noce river where we collected also touristic presences in the days of the sampling campaigns. However, we make a qualitative comparison of the spatial and seasonal patterns showed by the simulations with the observations after warning the reader of the time scale mismatch between simulations and measurements.

In his/her comments the reviewer focuses on specific aspects of the example that we used to illustrate a possible application of the model. This was very useful to us to identify points where our presentation needed an improvement. However, without reducing the importance of these valuable comments, we feel that the most relevant contribution of our manuscript is not the application per se, but the model we present which we believe contains elements of innovation worth to be discussed. We add a sentence in the presentation of the applications specifying that the simulations are intended to identify most impacted sites and seasonality of the concentration (in relative terms) rather than making an accurate prediction of the concentrations at the sampling points; which is the following: "It should be acknowledged that model's parameters are affected by uncertainty, which is expected to be large due to the limited number of data available for inference. For this reason the results of the simulations discussed here should be considered as a preliminary exploration providing uncertain estimates of concentrations at the sampling points. However, the simulated spatial pattern of the concentrations and its relative values are much more reliable than the absolute concentration at a given point, such that areas where the impact of PPCPs is stronger are reliably identified."

*Specific comments:*

*1. Existing knowledge: Concentrations of PPCPs*

*The paper lacks a description of existing studies on the occurrence and spatio-temporal dynamics of PPCPs in European rivers. In recent years several studies have been published in various river systems. This is important, because the results of the present studies should be compared to other regions to prove that both measured and simulated concentrations are realistic. I propose a table in the introduction where the five simulated compounds are selected and occurring concentrations are given for different rivers. Those should be used to evaluate model quality in the discussion. By the way: it is not true that PPCPs are not measured on a regular basis by environmental agencies: At least in the Rhine River measurement stations exist for this purpose (e.g. in Weil, check at: http://www.aue.bs.ch/umweltanalytik/rheinueberwachungsstation-weilam- rhein.html*

**Reply**

We partially agree with this comment. From one side including a table with examples of concentrations measured worldwide may be useful to evidence the relevance of the issue and the need for robust modelling approaches. On the other hand, we disagree with the comment that concentrations measured in other rivers may validate the measurements we used in the present work, since they depend on the amount of population, type of the water treatment and other factors that are site (catchment) specific. The quality of the measurements we used in the present work is granted by the protocols used in the sampling campaign, the care in maintaining and shipping the samples, and the analytical methodologies used in the lab. All this is discussed in previously published papers and therefore we add a sentence at page 12 and lines 11-14, specifying the care with which all the operations have been performed referencing to the work by Mandaric et al. (2017) for details. As for the sentence "PPCPs … are not monitored on a regular basis by environmental agencies" we were meaning that a regular monitoring is not yet enforced by regulations at the European level. This does not prevent agencies of member states to introduce independent monitoring activities, continuous or not, for specific compounds. To accommodate for the above comments we add the table suggested by the reviewer in the Introduction section (at page 5 of the revised manuscript) and the following sentences: 1) "The quality of the measurements utilized in the present work is granted by the protocols used in the sampling campaign, the care in maintaining and shipping the samples, and the analytical methodologies used in the laboratory. For further details on the protocols followed in sampling, handling, shipping and analyzing the samples, we refer to the previous work of Mandaric et al. (2017)" at page 12 and lines 11-14; and the sentence 2) "Despite these substances are ubiquitous in populated areas and detected in fresh waters with concentrations ranging from nanograms to micrograms per litre, a regular monitoring activity by Environmental Agencies is not yet enforced by regulations at the European level" at page 1 and lines 15-18 of the revised manuscript.

*2. Existing knowledge: Model approaches*

*The literature overview at the end concentrates on two, relatively similar GIS-based model approaches, GREAT-ER and PhATE, that are both more than ten years old. More recent approaches are missing, e.g. the study of Osorio et al (2012) who applied a simple plug-flow model to simulate pharmacologically active compounds in the Ebro river taking into account different dilution by varying flow conditions and an overall decay constant. They also determined decay constants for ibuprofen, furosemide, enrofloxacin, enalapril, acetaminophen, diclofenac and ketoprofen. So the authors should state why they developed a new model and what are the differences and benefits of their approach. Finally they should also compare their calibrated decay constants to those by (Osorio et al. 2012).*

**Reply**

We thank the reviewer for this comment. We considered PhATE and GREAT-ER as the state of the art because, to our best knowledge, they are the most comprehensive and used models available in literature. In addition, they have been applied and developed in recent years by several authors (among others, Aldekoa et al., 2015 for PhaTE and Kehrein et al., 2015 for GREAT-ER). Other models (e.g., Scheytt et al., 2006; Morales et al., 2007; Einsiedl et al., 2010; Vulliet et al., 2011; Osorio et al., 2012, Vione et al., 2018) share with GREAT-ER the conceptual framework, which is less general with respect to our model, particularly for what concerns the hydrodynamic processes. For example, the model presented by Osorio et al. (2012) does not take into account the change of velocity and the increase of dilution along the path from the source to the detection section. This is evident in the structure of the equations (4) and (6) of the paper by Osorio et al. (2012), which are used to estimate the residence time of the path from the source to the control section by using a single constant water discharge. This is legitimate if the distance from the source to the control section is such as the water discharge (i.e. the contributing area) changes little along the path, as admitted in the lines following equation (2) of the paper by Osorio et al. (2012). However, the validity of this approximation can be questioned in most applications dealing with multiple sources on a network. In Osorio et al. (2012) the effect of multiple sources is taken into account by rescaling the residence time with the annual volume of the sources as shown in equation (5). This rescaling is empirical. In our model we take into account both effects in a rigorous manner. Stream velocity changes stepwise along the network as the result of considering the water discharge constant along the single reach but changing from node to node according to the contributing area. In this framework dilution along the path is considered by performing mass balance at the nodes of the network. The effect of multiple sources is addressed in a rigorous manner by taking advantage of the linearity of the transport equation (see the presentation of the model). We believe that these characteristics are a significant improvement of the existing modelling approaches, including that referenced by the reviewer. Notice that neglecting dilution along the path may result in a severe overestimation of the concentration or, in case of calibration with observed concentrations, in a decay coefficient larger than the real one to compensate for unmodeled dilution. All these processes, not considered in previous approaches, are introduced in a simple, yet rigorous, manner in our modelling approach. We expand the discussion of the advantages of our approach with respect to the existing ones in the introduction section (from line 5 of page 3) of the revised manuscript.

*3. Missing information about model boundary conditions*

*How was the weather (rainfall, air/water temperature, etc.) during the two sampling periods? Since water temperature and discharge are two main parameters that influence the model results, this information is crucial to evaluate the model results. If figure 3 of Mandaric et al. (2017) is considered, the runoff difference between the sampling campaigns of February and July is rather small for the two downstream*

*sites WB4B and WB5B. Here absolute numbers should be presented in a table, in addition to weather conditions and air/water temperatures.*

**Reply**

We comply with the reviewer's request of providing information on meteorological and hydrological conditions during the sampling campaigns. Hence, we introduce a table with the available meteorological and hydrological data (i.e. water temperature and streamflow) in the subsection 3.2. For what concern rainfall data, we specify that the first sampling campaign was conducted in the dry season (winter) and the second in the period of snow melting (summer). However, in both campaigns no significant rainfall was detected during the sampling operations and in the previous days. We remark here that the model is structured in such a way that the effects of water discharge and temperature are automatically considered in the simulations. All the available information of measured data were already published as supplementary data in the Appendix of Mandaric et al. (2017), to which we refer for further information.

*4. Missing information about Lake di Santa Giustina*

*A more accurate map of the study catchment is required. Unlike in the preceding paper, an existing large lake between upstream (WB3A and WB3B) and downstream (WB4B and WB5B) sites is not shown in the catchment map and even not mentioned in the paper. Why? This lake of 180 million m3 volume will largely increase residence times and mixing and certainly affect model results. This could also explain differing model results between the upstream and downstream stations. Moreover, such a large lake has impacts on various simplifying model assumptions, as discussed below. Here an extensive discussion is needed including a quantitative estimate on the solute residence times in the lake (at different discharge conditions).*

**Reply**

We thank the reviewer for noticing this. We change the figure as required. Concerning the second, more important, comment one may note that in principle natural lakes and reservoirs can be easily implemented in our modelling framework by adding the pdf of its residence time into equation (12). However, from the application point of view, the selection of a proper residence time pdf requires information on hydrodynamic characteristics such as the distribution of the residence times and the type of micro-mixing, which we did not find at a first inspection of available information on the S. Giustina reservoir. Since we agree with the reviewer on the importance of the reservoir for the downstream sampling points, we performed a throughout analysis of available reports and elaborated a simplified, yet accurate, model of the reservoir. A first consideration is that the reservoir is in between sections WB3B and WB4B and shows a significant increase of the contributing area with respect to the former section. In the current approach we introduce the hypothesis of perfect mixing, which entails that the outflows sample ages proportionally the volume-weighted distribution of ages available in the reservoir. These changes

required to perform again both the calibration and the simulations over the entire Adige river. To comply, and we think to resolve the issue raised by the review, we make the following changes to the original manuscript: 1) modify the description of the Adige catchment in the subsection 3.1 (from line 24 of page 10 to line 13 of page 11) by adding a detailed description of the reservoir and the residence times as follows: "The Noce river is exploited for hydropower production with 4 reservoirs, two in the upper course (Careser and Pian Palù) and two in the middle course (S. Giustina and Mollaro). Careser and Pian Palù are in headwaters with no WWTPs upstream and therefore they enter in the model only with their effect on the water discharge. The other two are downstream of a few WWTPs (see Figure 2 and Table B1 in the Appendix B) and therefore their effect on the residence time is included. The Mollaro reservoir is just downstream of the S. Giustina reservoir and since no release points are in between we merged them in a single equivalent reservoir. A recent publication of the Hydrological Observatory of the Trento Province (2007) shows that in the period 2001-2005 the average volume stored in the S. Giustina reservoir was of $12.089 \cdot 10^6$ m$^3$. In the same period the mean water discharge was 25.8 m$^3$ s$^{-1}$, thereby leading to a mean residence time of $\tau_{s_1} = \frac{\bar{V}}{\bar{Q}} = 53.7 \ days$. Mollaro has a little storage volume compared to that of S. Giustina. At the maximum storage (i.e., $0.860 \cdot 10^6$ m$^3$) the mean residence time is of $\tau_{s_2} = 0.38 \ days$, which summed to the mean residence time of S. Giustina leads to a total residence time of the two reservoirs of $\tau_{s=}\tau_{s_1} + \tau_{s_2} = 54$ days. Notice that the storage of Mollaro has been considered constant because of its small volume which allows very little flexibility for storing the water released from the S. Giustina reservoir (the S. Giustina reservoir feeds the Taio power plant which release point is just upstream of the Mollaro reservoir) and counts only for a small fraction of the total residence time of the two reservoirs. In this situation the water coming from S. Giustina and the small catchment between the two reservoirs is stored for a very short time in the Mollaro reservoir with respect to the residence time of S. Giustina, such that fluctuations of its storage volume are not influencing significantly $\tau_s$"; 2) modify Figure 2 according to reviewer's suggestion; 3) modify equation (12) at page 7 to include the effect of lakes and reservoirs (namely by adding the pdfs of the lakes to the convolution) and by adding the following text at page 9 and lines 25-28, after equation (15): "If a lake is encountered along the path and its functioning can be represented as a plug-flow such that $g_M(s_k, t) = \delta(t - \tau_{s_k})$, Eq. (15) should be generalized by adding the residence time $\tau_{s_k}$ (and that of the other lakes encountered along the path) to the channels residence times $\tau_{i,j}$. If the hypothesis of plug-flow does not hold, the pdfs of all the lakes encountered along the path should be convoluted to the Eq. (15)".

**5. Model assumptions**

*Limitations caused by the following model assumption need adequate discussion: a.) the model only accounts for point sources. How large is the error by diffuse sources, e.g. sewer overflows during large rainfall events, manure use, etc. b.) the entire*

*concept is based on the hypothesis that the scale of interest is larger than the residence time. Does this hold, especially regarding the big lake? c) can velocity be assumed to be spatially uniform within the channels (p 4)? Generally it seems more plausible that flow velocities are higher in steep headwaters and again, what about the lake? d) also the assumption that local dispersion is overwhelmed by geomorphological dispersion is problematic if one thinks about a large lake. e) how realistic is the assumption that every WWTP is equally efficient, i.e. has the same decay factor (p. 10)? From my experience there are large differences depending on the size of the WWTP, on its age and used techniques. Snip et al. (2014) exemplified the effect of different operation conditions on WWTP efficiency regarding PPCP removal. f) was the water temperature really constant throughout the network and throughout the three-day sampling campaign? Here again the balancing effect of the lake might play a role. Real measured data needs to be presented here (see 3. above).*

**Reply**

We thank the reviewer for these comments. We will answer to each point in the same order.

a) Our model is based on a segmentation of the path from the source to the control section with the source that is introduced at the location where the WWTP discharges into the river. Therefore, diffused contributions can be evaluated at the level of the sub-catchment draining into the reach and treated as a point source located at its middle point. The details to which the variability of the diffused contribution is considered is controlled with the drainage density (i.e. the density of the network) and in many cases this is a better approximation of the diffused contribution than assuming (arbitrarily) a uniform distribution, which however can be handled if needed simply by assuming the same diffused specific load in each sub-catchment. We comment this point in section 2.1 of the revised manuscript (page 9, lines 4-11) as follows:

"Notice that the model is based on a segmentation of the path from the source to the control section. Therefore, diffused contributions can be evaluated at the level of the sub-catchment and treated as a point source located at the middle of the channel draining the sub-catchment. The length of the channels composing the river network, and therefore the size of the sub-catchments can be varied according to the threshold in the contributing area chosen during the identification of the river network from the Digital Terrain Model (see e.g. Rodriguez-Iturbe and Rinaldo, 1997). Hence, the maximum detail with which the spatial variability of the diffused contribution is reproduced can be controlled by the modeler simply by changing the density of the network. Notice that this has also an effect on the minimum time scale at which variability of the flow field can be captured, through the control of the channel length on the residence time."

For the case at hand, sources of diffused origin are not relevant since the region uses separate sewer systems, which eliminate sewer's overflow and the possible input from manure cannot be evaluated with the information available. We add the following caveat in Section 4 (from line 31 of page 13 to line 1 of page 14): "To comply with Occam's razor principle (MacMay, 2003 ch. 28), suggesting parsimony in selecting model complexity and considering the very limited amount of concentration data available, the parameters in the Eq. (13) are assumed the same for all the WWTPs and since $\gamma$ is inferred the abatement $f$ is assumed to be zero. This simplification is supported by the fact that all WWTPs of each of the two provinces are managed by the same agency by using similar technologies".

b) This is not what we meant and also in this case one should separate the characteristics of the models, and the processes it includes, with the approximations introduced for the case at hand. The approximation introduced applies to equation (2), which refers to the single reach not to the entire pathway as the reviewer implicitly assumes. In this context what we require is that the residence time in the single reach is smaller than the characteristic time of change of water discharge. In a single reach the residence time may vary between minutes to hours, thereby the approach may be questioned only during flooding events. We are currently exploring what we "loose" by using this parsimonious approach also during flooding events, but discussing this is beyond the scope of the present work. At the level of the entire network the effect of a water discharge changing in time is addressed by means of equation (10). In the application, we introduced the further hypothesis that the ratio of water discharges in equation (10) can be approximated as the ratio of the contributing areas, which is a reasonably assumption in normal hydrological conditions (no flooding) as actually was the case during the sampling campaign. This is an approximation that is instrumental to simplify the computations given the type of data available but can be removed in applications if the data available allow to do it. For what concerns the reservoir the assumption is that fluctuations of the water discharge influence little the residence time, which depends on the ratio between the mean stored volume and the mean water discharge at the monthly or seasonal scale, depending on the data available. Again, our modelling approach is flexible enough to tailor the complexity of the model to available data and modelling goals. We are not saying here that modelling should be adapted to data available, rather that there should be a balance between modelling goals investment in data collection and modelling effort. We introduce in section 2 at page 4, lines 17-21, a caveat to better specify this point: "The model assumes that the velocity is steady state, but it can be used to simulate representative states of a slowly variable flow approximated as the superimposition of a sequence of steady state velocity fields. This is acceptable if the characteristic time of water discharge variations is larger than

the residence time within the channel. Considering the typical length of the channels composing a river network and the time scales at which the PPCPs loads are available, time variability can be captured at daily or larger time scales".

c) In equation (2), $v$ is the mean channel velocity and the effect of spatial variability of point velocity within the channel is taken into account by the diffusivity $\alpha_L$ introduced in the pdf (equation 6) of the travel time. It has been shown in the past (Rinaldo et al., 1991) that at the network level the effect of $\alpha_L$ on the shape of the hydrological response is negligible compared to the morphological dispersion reflecting dilution occurring at the nodes of the network and this justifies the approximation of $\alpha_L \rightarrow 0$ introduced in the equation (15). Notice that the velocity is spatially constant within the single reach, but varies along the network according to the scaling law introduced by Leopold (1953) and recently elaborated further by Dodov and Foufoula-Georgiou (2004) (see equation 3 and appendix A). According to these scaling laws, the velocity varies slightly along the network and increases moving downstream, as the contributing area increases, rather than reducing as argued by the reviewer. This is a well know result of early studies on river morphology that our model takes into account. A lake or a reservoir is a storage unit and therefore its functioning is different from that of a river. In particular, the reservoir is modelled with a transfer function that depends in a complex manner from reservoir hydrodynamics, which is typically epitomized through a relationship relating residence time and stored volume. This differs from a reach, where kinematic mechanisms, epitomized through the velocity, prevail over storage mechanisms. We specify how we implemented the reservoirs in the subsection 2.1 from the page 7 to 9 (see the reply to comment 4).

d) We do not understand this point. Local dispersion refers to streams, while the pdf of the residence time in lakes or reservoirs enters in the definition of geomorphological dispersion by increasing it. So the presence of the reservoirs makes this assumption even more accurate than in case of a network without them.

e) As before, we divide the answer to this question into two parts; what the model can do and what the available data actually permit. The efficiency of the WWTPs is included in our model through the decay coefficient $f_i$, which is compound specific and may change depending on the treatment technology. In the application we do not estimate $f_i$ just because it enters into the parameter $\gamma_i$ (see equation (13)), which is calibrated after assuming that all the WWTPs have the same efficiency. This is acceptable for the case at hand since all the WWTPs of the province of Trento are managed by the same public authority, which guarantees uniformity in the conduction, and the employed technology is the same. Similar considerations can be made for the

WWTPs of the Bolzano province. We remark that the choice of the parameters that are not subjected to calibration (fixed) depends on the particular application and that in other situations, for example when $f_i$ is known, a different choice can be done. This is not by any means a limitation of the model, which as we said can handle the case of $f_i$ variable if needed and if proper data are available.

f) Yes water temperature is nearly constant, as can be seen in the Table 2 of the revised manuscript showing the relevant meteorological data during the sampling campaign (see the reply to the comment n. 3).

*6. Model check*

*Apart from the missing incorporation of the big lake, this is the main criticism of the presented approach: a missing model check with independent data. In the chapter 4 the model is extrapolated to the entire Adige river catchment and the spatial (Figure 5) and temporal (Figure 6) dynamics of selected components are shown. But how realistic are these results? Here the authors miss the chance of an independent model check. In their preceding paper Mandaric et al. (2017) presented additional data of the Adige catchment upstream the inflow of the Noce and downstream the town of Trento. This data could be used for a real model check and compared to simulated concentration WITHOUT re-calibrating the model.*

**Reply**

We thank the reviewer for this comment. Unfortunately we do not have enough data for a formal validation and, in addition, for sampling points along the main stem of the river Adige (WB6 and the four sites labelled WB7) census data of touristic presences are available only at the monthly time scale. However, these data allows a qualitative check based on the comparison of the spatial and temporal pattern seen in the data and obtained with the simulation. On the other hand a one to one comparison of the measured and simulated concentration may be misleading for the reason just discussed. To evidence this we add the following sentence at the beginning of section 5 (page 19, lines 16-20), specifying the use that can be done of the results of the simulations, which are intended to demonstrate a possible application of the model with the limitations due to the type of available data rather than the model itself: "It should be acknowledged that model's parameters are affected by uncertainty, which is expected to be large due to the limited number of data available for inference. For this reason the results of the simulations discussed here should be considered as a preliminary exploration providing uncertain estimates of concentrations at the sampling points. However, the simulated spatial pattern of the concentrations, and in particular their relative values, is much more reliable than the absolute concentration at a given point, such that areas where the impact of PPCPs is higher are reliably identified". In the same section, at page 21 and lines 3-10, we add the following sentences referring to the use of the additional information to which the reviewer refers in his/her appraisal: "During the two sampling campaigns samples were

collected and concentrations evaluated at the sites WB6, just upstream of the city of Trento and the confluence of both Noce and Avisio, and at four locations labelled WB7 A, B, C and D, downstream of Trento (see Mandaric et al. 2017 for location). This additional information cannot be used for a formal validation because representative of the sampling day, while simulations are conducted at the monthly scale because along the mean stem of the Adige river census data are available only at this scale for the touristic fluxes and at the annual scale for the resident population. While one can safely assume that resident population changes little within a year, touristic fluxes show significant variations at the weakly and even shorter time scales. Monthly concentrations produced by the model at selected sections are discussed below keeping in mind this limitation."

*7. Model description*

*A large part of the present manuscript consists of a presentation of the applied model approach, model selection and calibration strategy. The model description is lengthy and should be condensed. The section 2 "model" should be included into the section "material and methods". Here the model development itself should be described in a first subsection followed by parameterization and calibration strategies. Most of the equations should be moved into the annex, and figure 1 be omitted.*

**Reply**
We disagree with this reviewer's comment. Our intention is not here to present the data of a sampling campaign, and model them with an existing modelling approach, but rather to present a new model. With this objective in mind, we believe that the description of the model should be detailed enough to show the differences with existing approaches and discuss prons and cons of the proposed approach. Figure 1 is important to discuss the conceptual framework and therefore we decide to keep it.

*8. Code and data availability*

*The sampled data should be provided as a table in the annex (upon publication the data is already published twice). Also the model code could be provided as an executable file.*

**Reply**
We thank the reviewer for the comment. All sampled data were already available online in the Appendix of Mandaric et al. (2017) and are easily accessible (the publication is open access) and we think there is no need to replicate tables here. Again, our objective is not to publish data, which have been already published as mentioned by the reviewer, but to present a new modeling approach. The model code is available upon request as declared at the end of the revised manuscript.

*Technical corrections:*

*The entire manuscript needs a thorough language check by a native speaker. Internet-Links should be omitted from the main text.*

**Reply**

We thank the reviewer for the suggestion. We check carefully the English language and we omit internet-links from the main text.

**References included in the revised manuscript**

Aldekoa, J., Marcé, R., and Francés, F. (2015). Fate and Degradation of Emerging Contaminants in Rivers: Review of Existing Models, in: Emerging Contaminants in River Ecosystems, pp. 159–193, Springer.

Dodov, B. and Foufoula-Georgiou, E. (2004). Generalized hydraulic geometry: Derivation based on a multiscaling formalism, Water Resources Research, 40.

Einsiedl, F., Radke, M., & Maloszewski, P. (2010). Occurrence and transport of pharmaceuticals in a karst groundwater system affected by domestic wastewater treatment plants. Journal of Contaminant Hydrology, 117(1-4), 26-36.

Kehrein, N., Berlekamp, J., and Klasmeier, J. (2015) Modeling the fate of down-the-drain chemicals in whole watersheds: New version of the GREAT-ER software, Environmental Modelling & Software, 64, 1 – 8, https://doi.org/https://doi.org/10.1016/j.envsoft.2014.10.018.

Leopold, L. B. and Maddock, T. (1953). The hydraulic geometry of stream channels and some physiographic implications, vol. 252, US Government Printing Office.

Mandaric, L., Diamantini, E., Stella, E., Cano-Paoli, K., Valle-Sistac, J., Molins-Delgado, D., ... & Sabater, S. (2017). Contamination sources and distribution patterns of pharmaceuticals and personal care products in Alpine rivers strongly affected by tourism. Science of the Total Environment, 590, 484-494.

Morales, T., de Valderrama, I. F., Uriarte, J. A., Antigüedad, I., & Olazar, M. (2007). Predicting travel times and transport characterization in karst conduits by analyzing tracer-breakthrough curves. Journal of hydrology, 334(1-2), 183-198.

Osorio V. et al. (2012): Occurrence and modeling of pharmaceuticals on a sewage impacted Mediterranean river and their dynamics under different hydrological conditions, Science of the Total Environment 440, 3–13.

Rinaldo, A., Marani, A., and Rigon, R. (1991). Geomorphological dispersion, Water Resources Research, 27, 513–525.

Rodriguez-Iturbe, I. and Rinaldo, A. (1997). Fractal River Basins: Chance and Self-Organization, Cambridge University Press.

Scheytt, T. J., Mersmann, P., & Heberer, T. (2006). Mobility of pharmaceuticals carbamazepine, diclofenac, ibuprofen, and propyphenazone in miscible-displacement experiments. Journal of Contaminant Hydrology, 83(1-2), 53-69.

Vione, D., Encinas, A., Fabbri, D., & Calza, P. (2018). A model assessment of the potential of river water to induce the photochemical attenuation of pharmaceuticals downstream of a wastewater treatment plant (Guadiana River, Badajoz, Spain). Chemosphere, 198, 473-481.

Vulliet, E., & Cren-Olivé, C. (2011). Screening of pharmaceuticals and hormones at the regional scale, in surface and groundwaters intended to human consumption. Environmental pollution, 159(10), 2929-2934.

**Reply to Referee #2**

*COMMENTS FROM Referee #2*

*The manuscript describes a model of transport of contaminants in surface water. Specifically, five pharmaceuticals (diclofenac, ketoprofen, clarithromycin, sulfamethoxazole and Irbesartan) are modelled in a case study application. The proposed framework is original and the main innovative contribution is related to the stationarity hypothesis and to the number of parameters. The manuscript is well written and pleasant to read. The model is appropriately described and the case study is rich of information. Results are promising and the proposed model represents an interesting amelioration compared to the previous ones present in literature. I have only a minor suggestion, that is, to improve the figure 3 (first line: diclofenac) since apparently it is not clear that there are also blue points under the red ones.*

**Reply**

We thank the Reviewer for his/her appreciation of our work. We improve figure 3 by substituting the bullets with other symbols.

**A parsimonious transport model of emerging contaminants at the river network scale**

Elena Diamantini[1], Stefano Mallucci[2], and Alberto Bellin[1]

[1]Department of Civil, Environmental and Mechanical Engineering, University of Trento, via Mesiano 77, 38123 Trento (Italy)
[2]C3A - Center Agriculture Food Environment, University of Trento / Fondazione Edmund Mach, via Edmund Mach 1, 38010 San Michele all'Adige (Italy)

**Correspondence:** Elena Diamantini (elena.diamantini@gmail.com)

**Abstract.** Waters released from wastewater treatment plants (WWTPs) [..[1] ]represent a relevant source of pharmaceuticals and personal care products to the aquatic environment, since many of them are not effectively removed by the treatment [..[2] ]systems. The consumption of these products increased in the last decades and concerns have consequently risen about their possible adverse effects on the freshwater ecosystem. In this study, we present a simple, yet effective, analytical model of transport of contaminants released in surface waters by WWTPs. Transport of dissolved species is modeled by solving the Advection-Dispersion-Reaction Equation (ADRE) along the river network by using a Lagrangian approach. We applied this model to concentration data of five pharmaceuticals: diclofenac, ketoprofen, clarithromycin, sulfamethoxazole and irbesartan collected [..[3] ]during two field campaigns, conducted in February and July 2015 [..[4] ]in the Adige river, North-East of Italy. The model showed a good agreement with measurements and the successive application at the monthly time scale highlighted significant variations of the load due to the interplay between streamflow seasonality and variation of the anthropogenic pressure, chiefly due to the variability of touristic fluxes. Since the data required by the model are widely available, our model is suitable to large-scale applications.

**1 Introduction**

The presence of pharmaceuticals and personal care products (PPCPs) in the environment raises growing concerns because of their potential harmful effects on humans and freshwater ecosystems (Ebele et al., 2017). Despite these substances are ubiquitous in populated areas and detected in fresh waters with concentrations ranging from nanograms to micrograms per litre [..[5] ](see e.g., Table 1), a regular monitoring activity by Environmental Agencies is not yet enforced by regulations at the European level (Heberer, 2002; Ellis, 2006; Kuster et al., 2008; Acuña et al., 2015; Rice and Westerhoff, 2017). The main entry route of PPCPs into the aquatic environment is through the water discharged by [..[6] ]Waste Water Treatment Plants
* * *
[1]removed: are
[2]removed: system
[3]removed: in
[4]removed: ,
[5]removed: , they are not monitored on regular basis

[revised manuscript text omitted]

---

## Author Response (AR2)

NOTE: *Comments from the Reviewers are in BLUE and ITALIC.*

**Reply to Referee #1**

*COMMENTS FROM Referee #1*

*I acknowledge the efforts of the authors to modify their approach and make it more flexible to also include the effects of lakes and reservoirs. This was a major flaw of the preceding approach and improves the general applicability of the model to other regions where lakes or reservoirs are present in the river network. Thus most of my concerns have been addressed in the revised manuscript. Two points remain, why I opt for moderate revisions.*

*a.) The assumption of complete mixing and a constant mean residence time maybe problematic in reservoirs that are used for hydropower generation. Here I would like to see an additional sentence in the manuscript that admits this constraint.*

*b.) I understand the point that outside the Noce river missing data prevents model application in a daily time step. However, this does not prevent a model check. The authors present simulated monthly mean concentrations (Figure 6) and hence the monthly regime of pharmaceutical concentrations. This regime can very well be checked by spot samples taken during the winter and summer season the authors already published in their preceding paper (Mandaric et al 2017). The measured concentrations can e.g. be included into figure 6, when model simulations are performed for the catchments of sampling locations 7A-D of (Mandaric et al. 2017). This would prove that the monthly regime is adequately represented by the model. And if not, deviations could be discussed.*

*Moreover I strongly recommend a thorough language check of the entire manuscript. I give one sentence (pp25-26) as an example:*

*"This additional information cannot be used for a formal validation because representative of the sampling day, while simulations are conducted at the monthly scale because along the main stem of the Adige river census data are available only at this scale for the touristic fluxes and at the annual scale for the resident population."*

**Reply**
We are glad to have fulfilled most of the concerns of reviewer 1. Below we reply point by point to the reviewer's remaining concerns and illustrate the changes implemented in the revised manuscript.
**concern a)** "*The assumption of complete mixing and a constant mean residence time maybe problematic in reservoirs that are used for hydropower generation. Here I*

*would like to see an additional sentence in the manuscript that admits this constraint*": we have already addressed this issue raised by the reviewer at lines 21-23 of page 7 and lines 18-21 of page 9 of the previous version by generalizing the model such that lake dynamics can be included through an appropriate selection of a travel time pdf or model of mixing. The choice depends on the information available on lake (reservoir) dynamics, the purpose of the simulations and the amount of information available. In the case at hand, we were able to obtain a reliable estimate of the mean renewal time of the S. Giustina reservoir, which is an important indicator of reservoir's dynamics. Additional information such as the distribution of the velocity and the local diffusion coefficient cannot be estimated and this prevents the identification of the 'exact' travel time pdf. On the other hand, the common assumptions of piston flow (with the travel time pdf equal to the Dirac delta function) or complete mixing (with an exponential travel time pdf) are not reliable as well. Notice that assuming an exponential pdf entails complete mixing in the sense that the probability for a particle to exit the reservoir is independent of its age, therefore a particle that just entered the reservoir has the same probability of exiting than all the other particles that entered before and are still in the reservoir. Saying this differently the particle mixes immediately with the entire reservoir's volume. If this is the complete mixing model the reviewer has in mind, we remark that indeed we did not use it. What we adopted is a model that assume local complete mixing within a volume equal to the incoming water volume in the previous 54 days. This is not complete mixing because the volume of mixing changes with the seasons depending on water discharge of the inflowing river. Saying this differently, we assumed that a particle that exits at a given time from the reservoir mixed with all the particles that entered in the 54 days before. This model is in between the two end members of piston and full mixing (i.e., exponential pdf) and we believe is a realistic, though simplified, model of the effect of this reservoir on the concentration of contaminants. We remark that the transport model per se is not limited to this model of mixing, since different mixing models can be accommodated, depending on the characteristics of the lake or reservoir, purpose of the simulations and data available, as we said at the beginning of this reply.

We reviewed and slightly modified the part of the manuscript dealing with the choice of the model of local mixing ( see lines 19-21 at page 13).

**concern b**) "*I understand the point that outside the Noce river missing data prevents model application in a daily time step. However, this does not prevent a model check. The authors present simulated monthly mean concentrations (Figure 6) and hence the monthly regime of pharmaceutical concentrations. This regime can very well be checked by spot samples taken during the winter and summer season the authors already published in their preceding paper (Mandaric et al 2017). The measured concentrations can e.g. be included into figure 6, when model simulations are performed for the catchments of sampling locations 7A-D of (Mandaric et al. 2017). This would prove that the monthly regime is adequately represented by the model. And if not, deviations could be discussed*":

as explained in the previous reply to this issue, a model check can be done if there are data available at the same time scale of the simulation. The comparison of data at a time scale smaller than the time scale of the simulations does not permit to identify if the differences are due modeling errors or small scale fluctuations (i.e. at scales smaller than the simulation time scale, yet larger that the observational time scale). This is our case in which significant concentration fluctuations occurs at the daily scale due to fluctuations in the releases.

These two effects cannot be disentangled and therefore, differently from the reviewer, we believe that a grounded discussion cannot be performed with these data. In the previous manuscript we already added words of cautions concerning the specific monthly simulations (see lines 6-9 at page 19 and lines 3-10 at page 20), and we believe that this is all that can be done with the available data. However, this does not diminish the relevance of our contribution, which is chiefly in the new modeling framework, which allows taking into account time variability of contaminant load and dilution due to spatial and temporal changes of water discharge.

To conclude, we checked again the language of the manuscript as suggested by the reviewer.

[revised manuscript text omitted]